# Potential of fecal microbiota for early-stage detection of colorectal cancer

Georg Zeller[1,†], Julien Tap[1,2,†], Anita Y Voigt[1,3,4,5,†], Shinichi Sunagawa[1], Jens Roat Kultima[1], Paul I Costea[1], Aurélien Amiot[2], Jürgen Böhm[6,7], Francesco Brunetti[8], Nina Habermann[6,7], Rajna Hercog[9], Moritz Koch[10,‡], Alain Luciani[11], Daniel R Mende[1], Martin A Schneider[10], Petra Schrotz-King[6,7], Christophe Tournigand[12], Jeanne Tran Van Nhieu[13], Takuji Yamada[14], Jürgen Zimmermann[9], Vladimir Benes[9], Matthias Kloor[3,4,5], Cornelia M Ulrich[6,7,15], Magnus von Knebel Doeberitz[3,4,5], Iradj Sobhani[2,*] & Peer Bork[1,5,16,**]

## Abstract

Several bacterial species have been implicated in the development of colorectal carcinoma (CRC), but CRC-associated changes of fecal microbiota and their potential for cancer screening remain to be explored. Here, we used metagenomic sequencing of fecal samples to identify taxonomic markers that distinguished CRC patients from tumor-free controls in a study population of 156 participants. Accuracy of metagenomic CRC detection was similar to the standard fecal occult blood test (FOBT) and when both approaches were combined, sensitivity improved > 45% relative to the FOBT, while maintaining its specificity. Accuracy of metagenomic CRC detection did not differ significantly between early- and late-stage cancer and could be validated in independent patient and control populations (*N* = 335) from different countries. CRC-associated changes in the fecal microbiome at least partially reflected microbial community composition at the tumor itself, indicating that observed gene pool differences may reveal tumor-related host–microbe interactions. Indeed, we deduced a metabolic shift from fiber degradation in controls to utilization of host carbohydrates and amino acids in CRC patients, accompanied by an increase of lipopolysaccharide metabolism.

**Keywords** cancer screening; colorectal cancer; fecal biomarkers; human gut microbiome; metagenomics

**Subject Categories** Cancer; Systems Medicine

**Mol Syst Biol. (2014) 10: 766**

## Introduction

Colorectal carcinoma (CRC) is among the three most common cancers with more than 1.2 million new cases and about 600,000 deaths per year worldwide (Jemal *et al*, 2011). In most cases, initial genomic alterations, for example, in the *APC/Wnt* signaling pathway, cause hyperproliferation, which can lead to the formation of adenomas, and finally invasive carcinomas upon accumulation of further driver mutations (Cancer Genome Atlas Network, 2012; Vogelstein *et al*, 2013). If CRC is diagnosed early, when it is still localized (American Joint Committee on Cancer (AJCC) stages 0, I, or II), the 5-year survival rate is > 80%, but decreases to < 10% for late diagnosis of metastasized cancer (in AJCC stage IV) (O'Connell

1  Structural and Computational Biology Unit, European Molecular Biology Laboratory, Heidelberg, Germany
2  Department of Gastroenterology and LIC-EA4393-EC2M3, APHP and UPEC Université Paris-Est Créteil, Créteil, France
3  Department of Applied Tumor Biology, Institute of Pathology, University Hospital Heidelberg, Heidelberg, Germany
4  Clinical Cooperation Unit Applied Tumor Biology, German Cancer Research Center (DKFZ), Heidelberg, Germany
5  Molecular Medicine Partnership Unit (MMPU), University Hospital Heidelberg and European Molecular Biology Laboratory, Heidelberg, Germany
6  Division of Preventive Oncology, National Center for Tumor Diseases (NCT) Heidelberg, Heidelberg, Germany
7  German Cancer Research Center (DKFZ), Heidelberg, Germany
8  Department of Surgery, APHP and UPEC Université Paris-Est Créteil, Créteil, France
9  Genomics Core Facility, European Molecular Biology Laboratory, Heidelberg, Germany
10  Department of General, Visceral and Transplantation Surgery, University Hospital Heidelberg, Heidelberg, Germany
11  Department of Radiology, APHP and UPEC Université Paris-Est Créteil, Créteil, France
12  Department of Medical Oncology, APHP and UPEC Université Paris-Est Créteil, Créteil, France
13  Department of Pathology and LIC-EA4393-EC2M3, APHP and UPEC Université Paris-Est Créteil, Créteil, France
14  Department of Biological Information, Tokyo Institute of Technology, Tokyo, Japan
15  Fred Hutchinson Cancer Research Center (FHCRC), Seattle, WA, USA
16  Max Delbrück Centre for Molecular Medicine, Berlin, Germany
   *Corresponding author. Tel: +33 1 49814358; E-mail: iradj.sobhani@hmn.aphp.fr
   **Corresponding author. Tel: +49 6221 3878361; E-mail: bork@embl.de
   †These authors contributed equally to this work
   ‡Present address: Department of Abdominal, Thoracic and Vascular Surgery, University Hospital Carl Gustav Carus, Technical University Dresden, Dresden, Germany

*et al*, 2004). Therefore, population-wide screening and prevention programs are recommended in many countries. Fecal occult blood testing (Hemoccult FOBT) is currently the standard noninvasive screening test (Levin *et al*, 2008; Zavoral *et al*, 2009). However, because FOBT has limited sensitivity and specificity for CRC and does not reliably detect precancerous lesions (Allison *et al*, 1996; Faivre *et al*, 2004), there is an urgent demand for more accurate screening tests to identify patients who should undergo colonoscopy, which is considered the most effective diagnostic method (Levin *et al*, 2008).

As the majority of CRC cases are thought to be of a sporadic nature rather than due to inheritance (Lichtenstein *et al*, 2000), environmental risk factors have been investigated for a long time, but only recently have microbes colonizing the gut been considered as potential cancer-promoting factors. While in gastric, hepatic, and cervical cancers, a causal role is established for a single infectious agent in each case, namely *Helicobacter pylori*, hepatitis B virus, and human papillomaviruses, respectively (de Martel *et al*, 2012), in CRC, a variety of bacterial species and tumor-promoting virulence mechanisms have been investigated, mostly in cell lines and mouse models. For example, *Bacteroides fragilis* strains producing genotoxins (BFTs) can induce inflammation, leading to DNA damage in host cells (Wu *et al*, 2009; Goodwin *et al*, 2011); similarly, *Escherichia coli* strains harboring a genomic virulence island (*pks*) can cause DNA damage and chromosomal instability in the host (Cuevas-Ramos *et al*, 2010; Arthur *et al*, 2012), and very recently, *Fusobacterium nucleatum* strains were reported to promote carcinogenesis upon invasion of host cells (Kostic *et al*, 2013; Rubinstein *et al*, 2013). However, it remains unclear how many CRC cases can be attributed to each of these agents, how these exactly interact with the human host or the microbial community in the gut, and whether altered microbial abundances may provide a basis for an accurate CRC screening test.

Obtaining a comprehensive view of the microbial ecosystem in our gut—the microbiome—has become possible with high-throughput environmental sequencing techniques (Qin *et al*, 2010; Human Microbiome Project Consortium, 2012), and a number of reports have associated gut microbiota with diseases, such as obesity, type 2 diabetes, and atherosclerosis (e.g., Karlsson *et al*, 2013; Koeth *et al*, 2013; Le Chatelier *et al*, 2013; Qin *et al*, 2012; Turnbaugh *et al*, 2009). Several medium-scale studies recently characterized the microbiota of colonic tumor biopsies compared to healthy mucosa either by quantifying the 16S rRNA phylogenetic marker gene or by metatranscriptomic sequencing (Marchesi *et al*, 2011; Castellarin *et al*, 2012; Kostic *et al*, 2012; McCoy *et al*, 2013; Warren *et al*, 2013; Flanagan *et al*, 2014; Tahara *et al*, 2014). Even though these consistently documented an enrichment of members of the *Fusobacterium* genus, the relevance of these or other microbial agents for noninvasive CRC screening remains unclear.

Here, we systematically investigate the potential of fecal microbiota for noninvasive detection of colorectal cancer in several patient populations from different countries.

# Results

## The gut microbiome of a French CRC study population

To explore associations between the gut microbiome and colorectal carcinoma (CRC), we first analyzed fecal metagenomes from a popu-

lation of 156 participants recruited in France (study population F in the following, see Table 1, Supplementary Table S1 and Supplementary Dataset S1 for patient data), who underwent colonoscopy to either diagnose colorectal neoplasia in the form of adenoma(s) (polyps) or CRC, or confirm the absence of these. Carcinomas were further classified according to established staging systems (AJCC and TNM) (O'Connell *et al*, 2004). We first analyzed global community properties: While CRC-associated dysbiosis did not result in significant changes of microbial community diversity or richness (Supplementary Fig S1D and F), the distribution of enterotypes, as a descriptor of global community structure (Arumugam *et al*, 2011), varied slightly, but significantly between patient groups (Supplementary Fig S1A and B). We further observed significant differences in the abundance of specific taxa (Kultima *et al*, 2012) (Supplementary Fig S2). The gram-negative phyla of Fusobacteria and, to a lesser extent, Proteobacteria were significantly increased in CRC patients, whereas Actinobacteria were decreased. Bacteroidetes and Firmicutes were enriched and depleted, respectively, in CRC patients (consequently, also the ratio between these two phyla (Turnbaugh *et al*, 2006) differed significantly, see Supplementary Fig S1C).

When comparing patients with adenomas (of any size) to neoplasia-free controls and to CRC patients in terms of their microbiota composition, we found them to be almost indistinguishable from neoplasia-free participants (significant differences, except for the *Ruminococcus* genus, could not be detected). Additionally, many of the CRC-specific changes were seen in the comparison to both neoplasia-free participants and adenoma patients (Supplementary Fig S2). For subsequent analyses of the CRC-associated microbiota, we therefore included patients with small adenomas (diameter < 10 mm) in the control group, whereas large adenomas as clinically significant precursors of CRC were excluded from these comparisons as has been done in other CRC screening studies before (Imperiale *et al*, 2014).

## A metagenomic classifier for CRC

To explore the suitability of gut microbiota for CRC detection, we evaluated the predictive power of global measures of taxonomic community composition. While these differed significantly between CRC patients and controls (Supplementary Fig S1G–J), they would not allow for accurate CRC detection, as quantified by an area under the receiver operating characteristic (ROC) curve (AUC) of 0.73 (Supplementary Fig S1K). As individual species abundance differences were already more discriminative (AUCs up to 0.75; Supplementary Fig S3), we hypothesized that a combination of marker species would lead to improved detection accuracy. To this end, we adopted a classification methodology based on penalized linear models (see Materials and Methods) (Tibshirani, 1996). The consensus signature, extracted from an ensemble of classifiers (see Materials and Methods) that were trained on taxonomic abundance profiles, consisted of 22 species (Fig 1A). On average, > 51% of the total absolute weight of these classification models can be attributed to the abundance differences of only the four most discriminative species: two *Fusobacterium* species, *Porphyromonas asaccharolytica* and *Peptostreptococcus stomatis*, all of which are enriched in CRC (Fig 1A). Although it is not yet clear whether and which gut microbiota are causally involved in CRC, recent evidence suggests *Fusobacterium* species (in particular *F. nucleatum*) to be prevalent CRC-associated microbes (Castellarin *et al*, 2012; Kostic *et al*, 2012)

**Table 1.  Summary of study population F, G, and H.**

| Study population | Healthy | Adenoma | | Colorectal cancer | | | | | Country of residence |
|---|---|---|---|---|---|---|---|---|---|
| | | | | Early stages[c] | | | Late stage[c] | | |
| | | Small (< 1 cm) | Large (≥ 1 cm) | 0 | I | II | III | IV | |
| F (*N* = 156)[a] | 61 | 27 | 15 | 0 | 15 | 7 | 10 | 21 | France |
| G (*N* = 38)[a] | 0 | 0 | 0 | 25 | | | 13 | | Germany |
| H (*N* = 297)[b] | 297 | 0 | 0 | 0 | | | 0 | | Denmark[d], Spain[e], Germany |

[a]Disease status confirmed by colonoscopy.
[b]Absence of neoplasias not assessed by colonoscopy.
[c]AJCC staging.
[d]Published in Qin *et al* (2010) and Le Chatelier *et al* (2013).
[e]Published in Qin *et al* (2010).

and to accelerate tumorigenesis (Kostic *et al*, 2013; Rubinstein *et al*, 2013). Here, we further refine this association to the two subspecies *F. nucleatum vincentii* and *F. nucleatum animalis*, both of which are distinct enough from *F. nucleatum* subsp. *nucleatum* and from each other to qualify as independent species (Supplementary Fig S4A) (Mende *et al*, 2013). While *Fusobacterium* is a promising candidate for a causal agent, only classification models with additional species resulted in precise CRC detection (Supplementary Figs S1G–K, S3 and S4). When cross-validating the metagenomic classifiers on study population F, an AUC of 0.84 was achieved (Fig 1B).

Whereas cases and controls in this study population were characterized by a similar distribution of gender and body mass index (BMI), CRC patients were significantly older on average (Supplementary Table S1 and Supplementary Fig S5A–D). To investigate this potential source of confounding, we trained a CRC classifier based on patient characteristics (gender, age, and BMI) alone and found it to have a cross-validation accuracy of 0.63, significantly less than the metagenomic classifier (Supplementary Fig S5E). Additionally, we tested whether the metagenomic classifier exploits potential correlations of microbial abundances with host age rather than with CRC, which would result in spurious CRC predictions for older subjects. However, the classifier showed an increase neither in false-positive rate nor in sensitivity for older patients (Supplementary Fig S5F and G).

In our study population, the accuracy of the metagenomic classifier was slightly better than that of the Hemoccult FOBT (Fig 1B); this test, basically detecting traces of blood in feces, is routinely used in mass screening for CRC (Allison *et al*, 1996; Faivre *et al*, 2004; Zavoral *et al*, 2009) and was also applied to participants prior to colonoscopy. The accuracy of another experimental screening assay based on a different readout, namely the level of gene methylation of the *Wnt* signaling pathway member, *wif-1* (Lee *et al*, 2009; Mansour & Sobhani, 2009), was also matched by the metagenomic test (Fig 1B). As the features captured by FOBT and the metagenome appeared to be partially independent (Fig 1A), we also evaluated a combination test. This significantly improved accuracy to an AUC of 0.87, corresponding to a considerable relative gain in sensitivity (i.e., TPR) of > 45% over the FOBT alone (Fig 1B).

**Validation of the CRC classifier using patients and controls from different countries**

The broad utility of gut microbial markers might be limited by geographical or ethnical particularities or by technical variations in experimental procedures (Sunagawa *et al*, 2013; de Vos & Nieuwdorp, 2013), as illustrated by distinct microbial associations with type 2 diabetes in Chinese and Swedish studies (Qin *et al*, 2012; Karlsson *et al*, 2013). We therefore sought to validate the metagenomic CRC classifier in an independent group of individuals from different countries (see Materials and Methods). To assess whether it maintains high specificity in a large control population, we applied the classifier to five samples from healthy individuals living in Germany and 292 published fecal metagenomes from Danish and Spanish individuals who were diagnosed with neither CRC nor inflammatory bowel disease (IBD) (Qin *et al*, 2010; Le Chatelier *et al*, 2013) (study population H, see Table 1, Supplementary Table S1 and Supplementary Dataset S1). At a decision boundary of 0.275 (i.e., the value above which the classifier predicts CRC, Fig 1), the resulting false-positive rate (i.e., 1 − specificity) varied slightly from 8.0 to 7.7% between cross-validation on study population F and independent validation on study population H (Supplementary Fig S6A). In these validation populations, the true error rate might be slightly overestimated, because absence of CRC has not been confirmed by colonoscopy. To also independently validate the sensitivity of the classifier, we sequenced an additional 38 fecal metagenomes from individuals who were diagnosed with CRC by colonoscopy in a German hospital (study population G, see Table 1, Supplementary Table S1 and Supplementary Dataset S1). On this dataset, the sensitivity (i.e., true-positive rate) of the metagenomic CRC classifier was 52.6% compared to 58.5% observed in cross-validation (Supplementary Fig S6B). On the combined validation set of study populations F and H, the metagenomic CRC classifier achieved an accuracy of 0.85 AUC, virtually the same value obtained in cross-validation of population F (Fig 1B and C). Taken together, these results indicate that, despite differences between study populations in nationality and demographics, metagenomic CRC detection is possible with high accuracy, broadly applicable and robust to technical variation.

In order to assess the potential improvement of classification accuracy with a larger study population, we included the 38 CRC patients from study population G into study population F and applied the same modeling approach to train and cross-validate a more comprehensive classifier (see Materials and Methods). Its cross-validation accuracy increased to 0.90 AUC (Supplementary Fig S6C and G), which illustrates the future promise of large, multi-center cohort studies investigating the role of the microbiota in this disease.

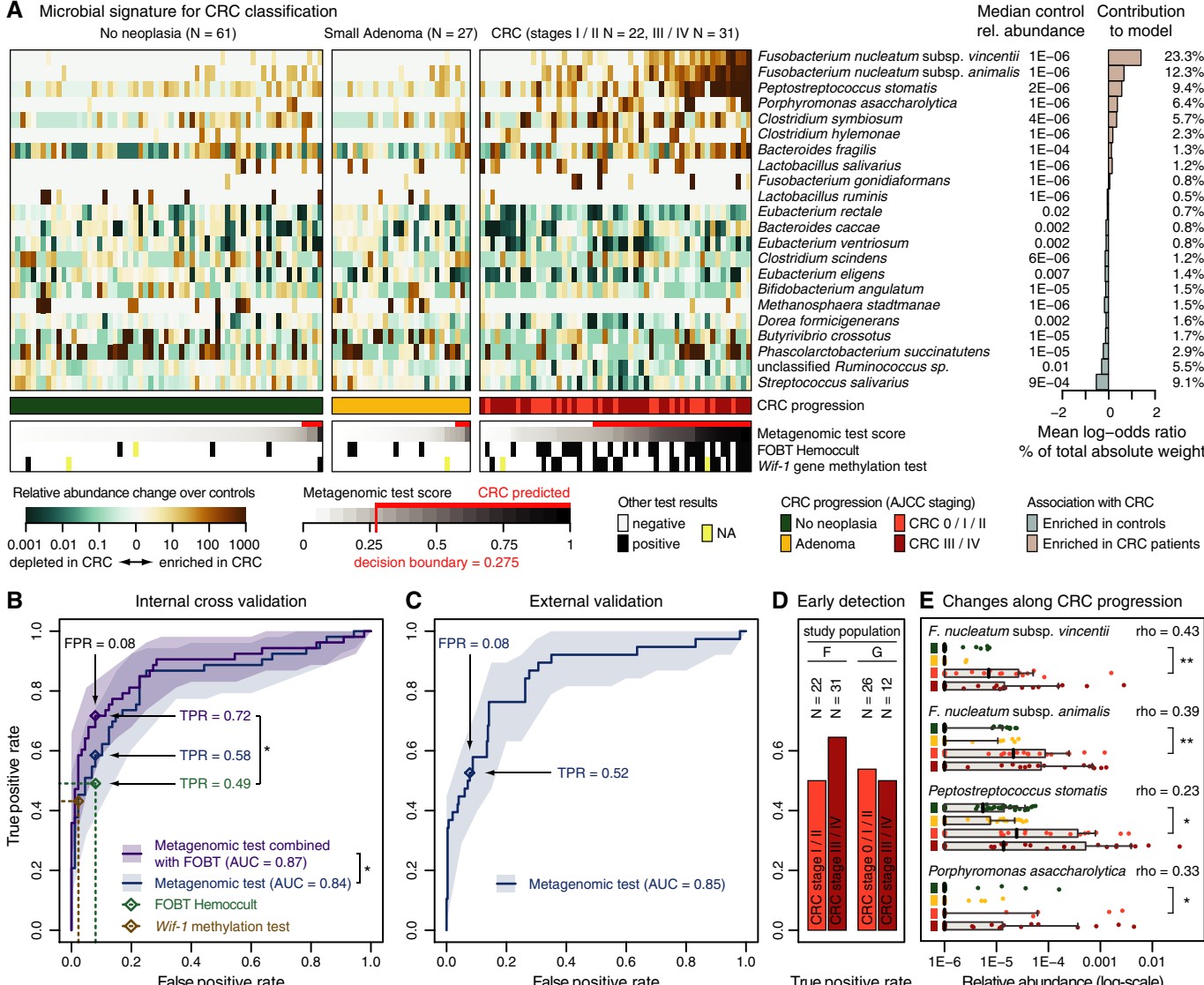

**Figure 1.**

## Detection of CRC in different stages of tumor progression

As an instrument for reducing CRC mortality, patient screening is most effective if cancer is diagnosed early, before the tumor has metastasized to nearby lymph nodes or distant tissues (O'Connell *et al*, 2004). To rule out that our metagenomic classifier is biased toward late-stage carcinomas, for which changes in the colon environment and its microbiota might be more pronounced than in earlier stages of tumor development, we compared the sensitivity for localized early-stage CRC (AJCC stages 0, I, and II) to that for metastasized late-stage tumors (AJCC stages III and IV). We found that early-stage CRC was detected with a sensitivity comparable to metastasized tumors in both study populations F and G (Fig 1D), confirming the potential of microbial markers for early detection of CRC. Additionally, we explored how the abundances of the four most discriminative CRC markers correlated with CRC progression from early neoplastic growth to late-stage metastasizing tumors (with samples stratified into four groups of neoplasia-free

participants, participants with adenomas, stage 0/I/II, and stage III/IV CRC patients) (Fig 1D). All of these markers showed a significant correlation (all Spearman *P*-values < 0.001), and a strong enrichment in early-stage CRC patients compared to controls was evident for both *Fusobacterium* species and *Peptostreptococcus stomatis* (Fig 1E). Likewise, depletion of some microbes in cancer patients (i.e., negatively associated with CRC), such as *Eubacterium* spp. *rectale* and *eligens* and *Streptococcus salivarius*, was already noticeable in early stages (Supplementary Fig S7).

## CRC marker species in inflammatory bowel disease

The observed associations of microbiota with CRC alone do not reveal how specific they are to this particular disease. To exclude that there might be a more general dysbiosis common to multiple disease conditions (e.g., due to inflammation), we applied our classifier to 25 recently published metagenomes from inflammatory bowel disease (IBD) patients (21 ulcerative colitis (UC) and four

◀

**Figure 1.  Signature of CRC-associated gut microbial species.**

A   Relative abundances of 22 gut microbial species, collectively associated with CRC, are displayed as heatmap in the left panel as fold change over the median relative abundance observed in controls (indicated to the right); the control group included neoplasia-free and small adenoma patients. The mean contribution of each marker species to the classification is shown to the right (bars correspond to log-odds ratio in logistic regression; numbers indicate percentage of absolute total weight, see Materials and Methods). Different cancer stages are color-coded below the heatmap (see Table 1, Supplementary Table S1 and Supplementary Dataset S1 for patient data). Below, the classification score of the microbial signature (from cross-validation) is shown as gray scale (see key) with the decision boundary and resulting false positives and true positives indicated in red (using colonoscopy results as a ground truth). Displayed alongside are the results of the standard Hemoccult FOBT routinely applied for CRC screening and an experimental CRC screening test based on methylation of the *wif-1* gene, a *Wnt* pathway member (Lee *et al*, 2009; Mansour & Sobhani, 2009) (see main text for details).

B   Test accuracy of the metagenomic classifier is depicted as ROC curve summarizing mean test predictions made in ten times resampled tenfold cross-validation on study population F ($N = 141$, 95% confidence intervals of true-positive rate are shaded, see Materials and Methods and Table 1). Additionally, the accuracy of the *wif-1* methylation test (Lee *et al*, 2009; Mansour & Sobhani, 2009) as well as of the FOBT is shown (as assessed for the same patients). A combination test, in which the FOBT results and microbial abundance profiles were jointly used as predictors, resulted in significantly enhanced accuracy over both the metagenomic classifier and the FOBT alone, compared to which the relative gain in sensitivity is > 45% at the same specificity (*denotes one-sided bootstrapping *P*-values < 0.05 of TPR improvement over FOBT and of difference in the whole ROC curve to the metagenomic test, respectively, see Materials and Methods). All screening tests are evaluated relative to colonoscopy findings (see key and main text for details; see also Supplementary Figs S4, S5 and S6 for additional details on the classifier, and Table 1, Supplementary Table S1 and Supplementary Dataset S1 for patient data).

C   When applied to the larger study populations G and H (335 metagenomes from several countries including 38 from German CRC patients) for external validation, the metagenomic classifier achieved very similar accuracy as in cross-validation, as measured by the area under the ROC curve (AUC) of mean test prediction scores (ROC curve and confidence intervals as in (A); see also Supplementary Figs S5 and S6 and Table 1, Supplementary Table S1 and Supplementary Dataset S1).

D   Sensitivity (TPR) of the metagenomic classifier for carcinomas in early stages (AJCC stages 0, I, and II) was similar as for late-stage, metastasizing CRC (AJCC stages III and IV) in both study populations F and G highlighting its potential utility for early detection (see also Table 1, Supplementary Table S1 and Supplementary Dataset S1).

E   Although the classifier associated species with a binary grouping into cancer and non-cancer patients, several of them exhibited gradual abundance changes over the progression from neoplasia-free participants over adenoma to early- and late-stage cancer patients (see key below A); displayed are the 4 most discriminative CRC marker species, each of which shows a Spearman correlation (rho) with cancer progression (grouped as in A) that is stronger than 0.2 with *P*-values < 0.001. Significant changes in early-stage CRC patients compared to neoplasia-free controls are marked (**P* < 0.05, ***P* < 1E-5, Wilcoxon test). Vertical black lines indicate median relative abundance with gray boxes denoting the inter-quartile range; gray whiskers extend to the 5th and 95th percentile (see also Supplementary Fig S7).

Source data are available online for this figure.

Crohn's disease (CD) patients, see Supplementary Dataset S1) (Qin *et al*, 2010). Although small sample size precludes precise estimates, our results indicate a moderate increase of false-positive predictions to 24% in these patients, about three times the rate seen in other controls (Fig 2A). This may reflect common alterations in the microbiota in these diseases, which is consistent with IBD patients being at greater risk of developing CRC (Bernstein *et al*, 2001). To explore this further, we monitored the four most discriminative CRC marker species for significant changes in abundance in IBD patients relative to controls (Fig 2B). Although higher prevalence of *Fusobacterium* species in IBD patients has been reported (Strauss *et al*, 2011), we did not observe a significant increase for the two CRC marker species from this genus in IBD patients, which is consistent with a recent study that showed an increase of *Fusobacterium* abundance in IBD patients' mucosal tissue, but did not detect an enrichment in stool (Gevers *et al*, 2014). For the most discriminative marker species that our model positively associated with CRC, we found significantly higher levels in CRC compared to IBD patients, indicating that these markers are specific to CRC (Fig 2B). Generally, stronger associations were observed with CRC than with IBD for most marker species (Supplementary Fig S8), suggesting that the metagenomic classifier is specific for CRC and is only modestly influenced by changes in the microbiota that are due to inflammation. However, to broadly compare changes in the gut microbiome across gastrointestinal disorders, larger studies on IBD and other inflammatory diseases will be needed.

**Fecal CRC markers reflect enrichments in tumor biopsies**

In order to make use of functional data extracted from the metagenome, fecal samples would need to reflect, at least partially, the microbial composition in the tumor environment. However, profiling colonic tissue samples with shotgun metagenomic sequencing is still ineffective due to excessive contamination with human DNA (Castellarin *et al*, 2012; Kostic *et al*, 2012). As an alternative, targeted sequencing of prokaryotic 16S rRNA gene (16S) fragments from tumor biopsies allows for taxonomic abundance estimation and identification of enriched microbes compared to nearby intact mucosa. For this, we newly sequenced 48 tumor–normal tissue pairs (13 from patients that were also part of study population G) and reanalyzed 79 such deeply sequenced pairs from a published study with US American, Vietnamese, and Spanish patients (Kostic *et al*, 2012) (see Supplementary Dataset S2 and Materials and Methods). To be able to distinguish relevant differences between the microbial communities at the tumor site and in stool from technical disparities due to different sample processing methods, we generated a third dataset of 116 fecal samples (part of study population F, see Supplementary Dataset S2) that were subjected to 16S amplicon sequencing. To profile the taxonomic composition of 16S samples, we constructed operational taxonomic units (OTUs, 98% sequence identity) from comprehensive databases of published 16S sequences (see Materials and Methods) and mapped 16S reads against these. To facilitate comparisons across datasets, OTUs were further matched to the marker species from the fecal metagenomic signature of study population F (using a best-hit approach based on the respective 16S fragments, see Materials and Methods). When comparing relative abundances between datasets (Fig 3), both fecal CRC marker species from the *Fusobacterium* genus showed a consistent enrichment at the tumor site, as was expected from previous studies (Castellarin *et al*, 2012; Kostic *et al*, 2012; Warren *et al*, 2013). Higher detection rates were observed in the tumor microenvironment compared to feces (Fig 3), suggesting a dilution effect that

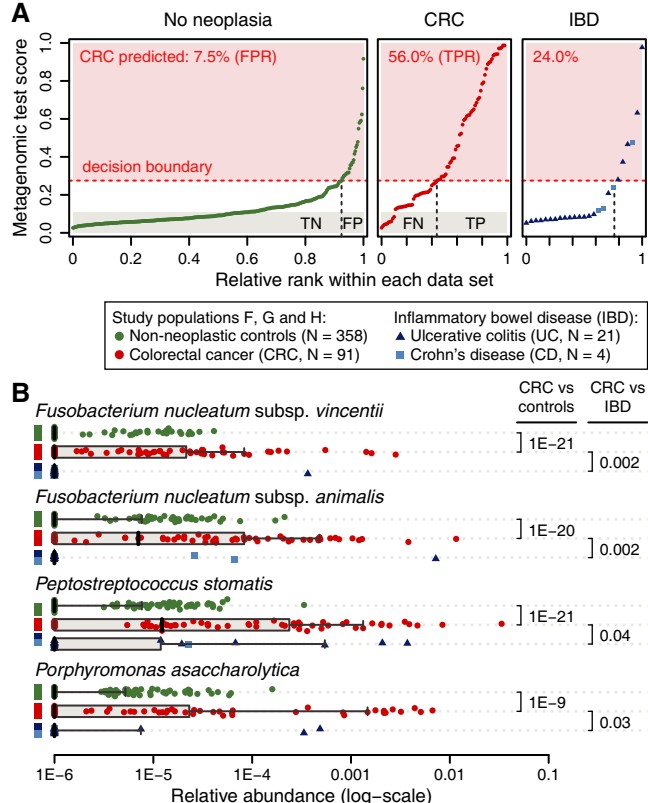

**Figure 2. Comparison of the CRC microbial signature to IBD microbiomes.**

A   Ranked predictions of the metagenomic classifier (mean test prediction for study populations F, G and H) are plotted for each individual (labels on top indicate grouping) with the percentage of positive CRC predictions annotated in red. Proportions of true negatives (TN), false positives (FP), false negatives (FN), and true positives (TP) are shown at the bottom for a decision boundary of 0.275 (see Fig 1). Application of the CRC classifier to metagenomes from ulcerative colitis (UC) and Crohn's disease (CD) patients indicates a threefold increased false-positive rate for inflammatory bowel disease (IBD) patients, suggesting some similarities between CRC and IBD.

B   Relative abundance distributions of the four most discriminative markers for CRC classification (see Fig 1A) are plotted for each patient subgroup, including UC and CD (see key). All markers showed significantly stronger association with (enrichment in) CRC than with IBD (Wilcoxon test, UC and CD tested together; see also Supplementary Fig S8). Boxplots are as defined in Fig 1E.

can, however, be overcome by deep sequencing of fecal material. Despite this, most of the abundance differences in feces between CRC patients and tumor-free controls were at least as significant as between tumor and normal tissue in the datasets compared here. We further observed increased abundance of the third CRC marker species, *Peptostreptococcus stomatis*, in CRC consistent with biopsies, but this trend was only significant in the published dataset (Kostic *et al*, 2012). Where comparability could be established across sequencing technologies, most metagenomic marker species with significantly decreased abundance in CRC patients from study population F also showed similar abundance changes in normal tissue compared to tumor, as it was the case for *Eubacterium* spp. and *Streptococcus salivarius* (Fig 3 and Supplementary Fig S9A). These results indicate that similar trends for the relative abundances

of marker species between fecal and biopsy samples from CRC patients are detectable despite the apparent differences in patient nationality, sample origin, experimental techniques, and analysis methodology. We furthermore verified that the similarity between tumor-associated microbiota and enrichments in fecal samples from CRC patients is not confined to the marker species only, but also manifests as a dominant trend in principal component analysis (PCA) (Supplementary Fig S9B). All this suggests that fecal readouts may also allow for inferences of the metabolic and functional potential of the colonic microbiome in the tumor environment.

## Functional changes in the CRC-associated fecal microbiome

To characterize microbial gene functions and how these differ between CRC patients and tumor-free participants, we quantified the relative abundances of prokaryotic KEGG (Kyoto Encyclopedia of Genes and Genomes) modules (Kanehisa *et al*, 2008) in each metagenome of study population F (see Materials and Methods). To investigate carbohydrate utilization preferences of the microbiota (Sonnenburg *et al*, 2005; Koropatkin *et al*, 2012; El Kaoutari *et al*, 2013), we additionally used prokaryotic families of carbohydrate-active enzymes from the CAZy database (Cantarel *et al*, 2009) to annotate metagenomes (see Materials and Methods). As a result, we found 24 KEGG modules and 20 CAZy families to significantly differ in abundance in CRC patients (Fig 4).

Analysis of the functions that significantly differed between healthy participants and cancer patients revealed a global metabolic shift from predominant utilization of dietary fiber in the tumor-free colon to more host-derived energy sources in CRC (Fig 4B). In healthy gut metagenomes, exclusively some fiber-degrading enzymes and fiber-binding domains were enriched, whereas in CRC metagenomes, the microbiota appeared to exploit growth substrates derived from host cells to a much larger extent (fivefold enrichment of host glycans in CRC, $P = 0.01$, Fisher test, Fig 4B). Thus, we hypothesize that an increased degradation of host glycans might be related to the etiology of CRC. However, because dietary data are not available for our study populations, we cannot rule out that differences in eating behavior between CRC patients and controls might contribute to the observed trends, as the gut metagenome can be affected by diet (e.g. Claesson *et al*, 2012; David *et al*, 2014; Le Chatelier *et al*, 2013; Turnbaugh *et al*, 2009). Host cell wall carbohydrates, such as mucins, whose utilization is enriched in the CRC metagenomes of study population F, have been established as an important energy source for commensal microbiota of the healthy gut (Sonnenburg *et al*, 2005; Koropatkin *et al*, 2012; Bergstrom & Xia, 2013). However, compromised integrity of the inner mucus layer that functions to shield the epithelium from luminal bacteria might accelerate the progression of CRC and inflammatory bowel disease: In animal models, mucin gene defects can lead to intestinal cancers or microbiota-dependent acute colitis (Velcich *et al*, 2002; Fu *et al*, 2011; Bergstrom & Xia, 2013). In IBD patients, increased mucolytic activity has been reported for mucus-associated bacteria (Png *et al*, 2010). It is therefore conceivable that degrading the mucus barrier might be a strategy that is adopted by adhesive and/or invasive pathogens such as *Fusobacterium* spp. to reach the epithelial cells (Dharmani *et al*, 2011; McGuckin *et al*, 2011; Rubinstein *et al*, 2013).

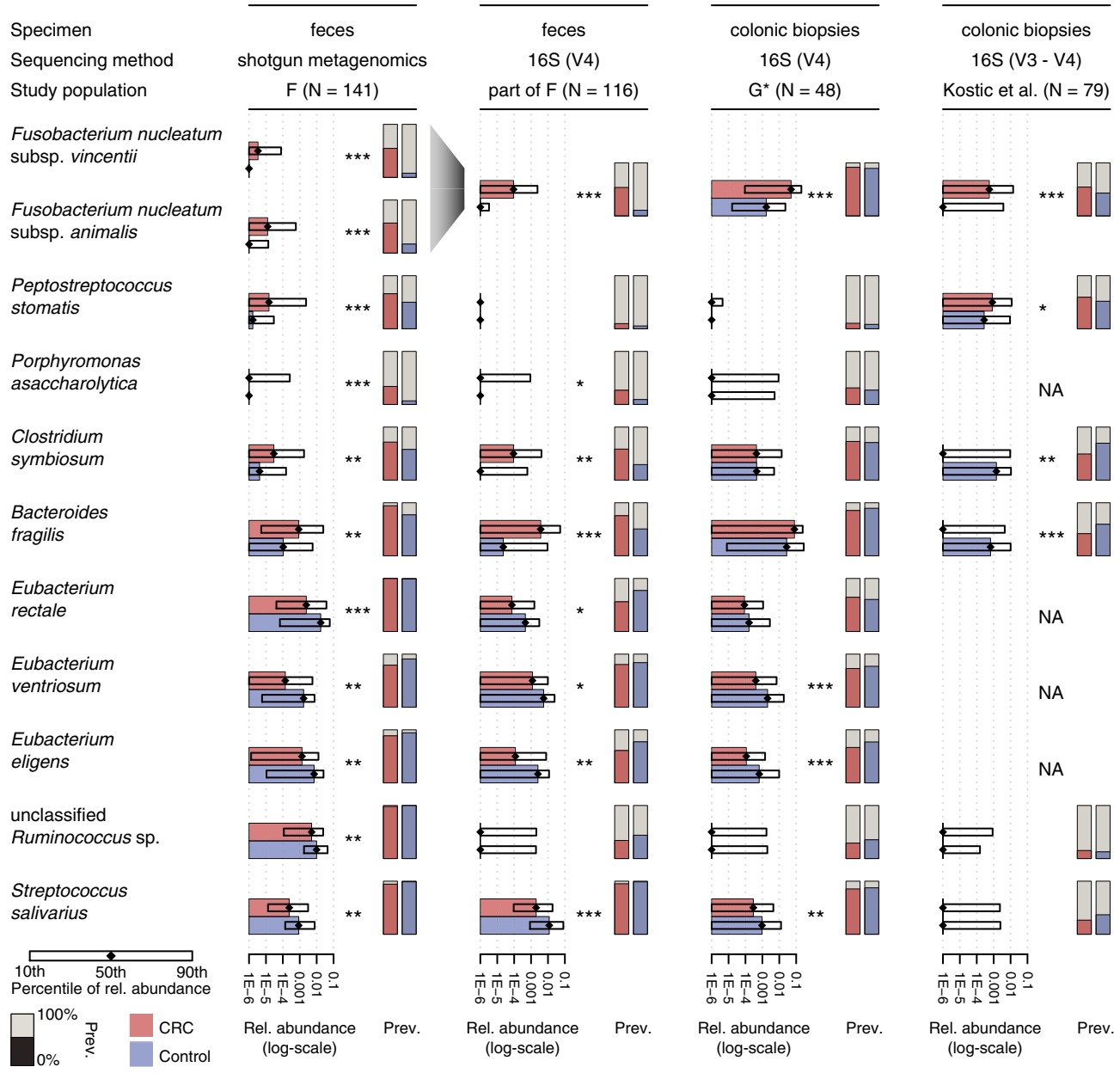

**Figure 3.  Consistency of CRC marker species abundances in fecal metagenomes and 16S rRNA profiles of tumor biopsies.**
Horizontal bars indicate changes in median relative abundance (rel. abundance) of the CRC marker species (as in Fig 1A) that significantly differed between CRC patients and tumor-free controls (excluding large adenomas; all nominal *P*-values < 0.005, Wilcoxon test, see also Supplementary Fig S9). These are compared to 16S OTU abundances from a subset of fecal samples from study population F as well as to two groups of patients in which microbial communities on tumor biopsies and healthy colonic mucosa were profiled and compared (of the 48 patients in study population G*, 13 are part of study population G, see also Kostic *et al* (2012)). Boxes denote the interval between the 10th and 90th percentile of relative abundance. Metagenomic marker species were matched to 16S OTUs using a best-hit approach for the amplified 16S rRNA gene regions (NA, not matched, see Methods), both *Fusobacterium* species were matched to the same 16S OTUs. Significance was assessed by unpaired and paired Wilcoxon tests for fecal and biopsy datasets, respectively (*nominal *P*-value < 0.05, **P*-value < 0.005, ***P*-value < 0.0005). Note that for the majority of the species shown, the significance for distinguishing CRC patients from controls is higher (lower *P*-value) in metagenomic than 16S readouts. Vertical bars display the prevalence (prev.) of CRC marker species per patient/sample group (percentage of individuals in which these species/OTUs were detected with a relative abundance exceeding 1E-5).

Some host cell-derived metabolites are more abundant in the tumor environment, for instance amino acids, of which elevated levels have been measured in CRC patients by metabolomics (Weir *et al*, 2013). Our data showed an increased capacity of the CRC-associated microbiota for uptake and metabolism of some amino acids via the putrefaction pathway (Fig 4A). The degradation products from this pathway include polyamines (like putrescine), which at increased intracellular levels promote tumor development (Gerner & Meyskens, 2004). It was recently shown that enterotoxigenic *Bacteroides fragilis* can exploit this by stimulating the endogenous polyamine catabolism in colonic epithelial cells (Goodwin *et al*, 2011). Although our functional analysis cannot reveal whether the

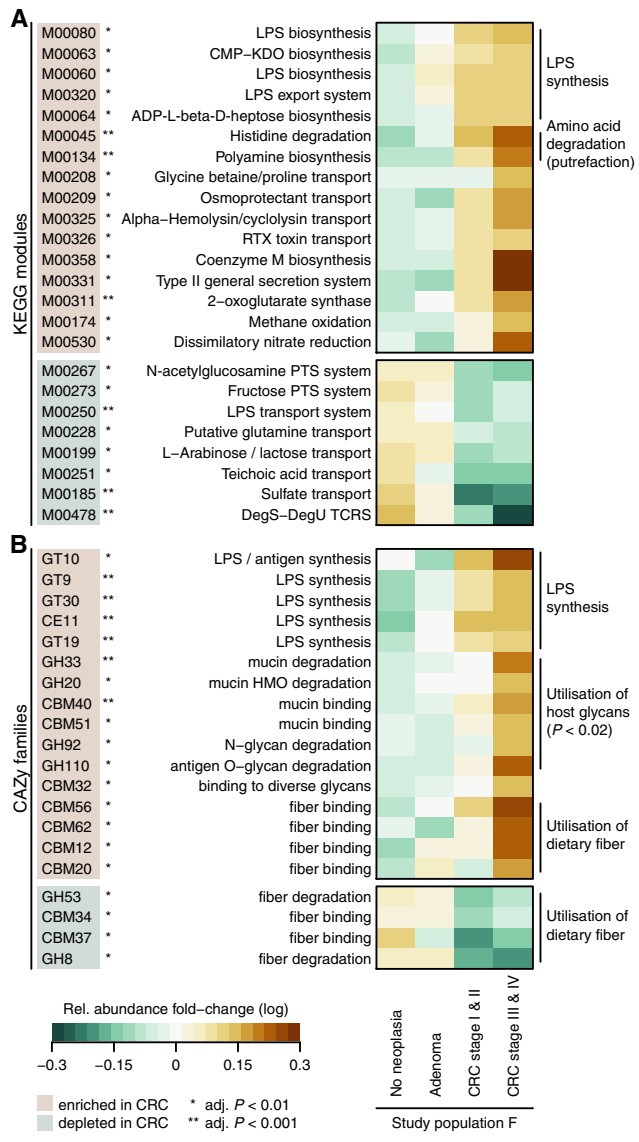

**Figure 4.  Functional changes derived from the CRC-associated metagenome.**

A   Significant changes in relative abundance of genes summarized by KEGG module annotations between cancer and non-cancer metagenomes are shown for cases with a > 1.33-fold change and an FDR-adjusted *P*-value < 0.01 (see legend and Materials and Methods). General trends in functional potential, such as enrichment of lipopolysaccharide (LPS) metabolism, and putrefaction in the CRC microbiome are summarized to the right of the heatmap.

B   Significant relative abundance changes of genes summarized by CAZy family annotation with a > 1.33-fold change and an FDR-adjusted *P*-value < 0.01 (see Materials and Methods). A metabolic switch to degradation of host carbohydrates, for example, mucins, in CRC metagenomes is annotated to the right. Moreover, a CRC-associated increase in metabolism of potentially pro-inflammatory bacterial cell wall components, such as lipopolysaccharide (LPS), is evident. Together with an increase of nitrate reduction in CRC metagenomes (A), this is consistent with a bloom of Proteobacteria (see also Supplementary Figs S2 and S3).

microbiome hypothesis' (Warren *et al*, 2013): Previously, it was noted that several CRC-associated bacteria, for example, *Fusobacterium* spp., were first described as oral pathogens, and it has been hypothesized that their invasion of the gut microbiome might cause or contribute to tumorigenesis (Warren *et al*, 2013). In line with this, *Peptostreptococcus stomatis* (Downes & Wade, 2006) and *Porphyromonas asaccharolytica*, which we found to be associated with CRC, were also described as oral pathogens before (Park *et al*, 2013). Similarly, putrescine/spermidine metabolism has been described as a core trait of the oral microbiota (Abubucker *et al*, 2012; Shafquat *et al*, 2014).

Concomitant with the metabolic shift in the CRC microbiome, we observed an expanded repertoire of pro-inflammatory and pathogenicity processes, most notably an increased potential for lipopolysaccharide (LPS) metabolism (Fig 4), which is consistent with a CRC-associated expansion of gram-negative bacteria that bear LPS antigens on their outer membranes. Through binding to Toll-like receptor 4 (TLR4) in epithelial cells, LPS triggers an inflammatory signaling cascade, which in turn could promote inflammation-induced carcinogenesis (Cario *et al*, 2000; Tang *et al*, 2010) and even metastasis, as has been demonstrated in mice (Hsu *et al*, 2011). An enrichment of hemolysin transport, RTX toxin transport, and type II secretion systems in CRC metagenomes hints at an increase of pathogenicity processes encoded in the genomes of gram-negative bacteria. To examine virulence factors and secreted toxins whose potential roles in the etiology of CRC were discussed before (Boleij & Tjalsma, 2012), we profiled a manually curated list of 15 toxin families and virulence factors in our metagenomic data. However, most of these (including *BFT* and the *pks* island) were either not detectable in fecal metagenomes or not enriched in CRC patients, except for *Fusobacterium* adhesin (*fadA*), which was recently shown to be required for its invasiveness and tumorigenesis (Supplementary Table S2; Rubinstein *et al*, 2013; Strauss *et al*, 2011).

Analyzed here for the first time, the complex functional alterations in the microbiome of CRC patients appear to occur gradually during CRC progression from precancerous stages to metastasized carcinomas (Fig 4). To directly assess this correlation, we applied PCA to significantly altered KEGG modules and CAZy families (as shown in Fig 4, using the same grouping of patients by CRC stage as above, see Materials and Methods). Indeed, the first principal component capturing the dominant source of functional variation between CRC patients and controls (PC1, explaining 43% of total variance) strongly correlated with neoplasia progression (Spearman's rho > 0.45, *P* < 1E-8, Fig 5), indicating that some functional changes in the microbiota are already detectable in early stages of neoplastic growth.

To explore whether also functional metagenomic profiles would be useful for CRC detection, we applied the above-described classification framework to KEGG module and CAZy family abundance data. Although the resulting models were less accurate (both AUCs 0.77) than the one based on marker species (Fig 1) and comparable to models utilizing taxonomic abundances summarized at higher ranks (Supplementary Fig S6D), a model based on both functional and taxonomic features yielded an improvement in accuracy over the best taxonomic model (AUC of 0.87 compared to 0.84; Supplementary Fig S6E and G).

observed enrichment of putrefaction in the CRC microbiome is a consequence of tumor metabolism or whether it contributes causally to tumor progression, it provides additional evidence for the 'oral

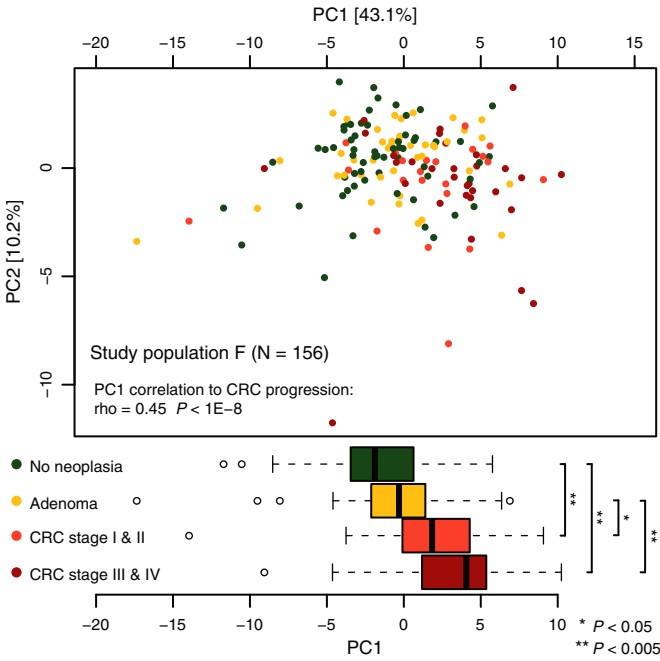

**Figure 5.  Functional changes correlate with CRC progression.**
Principal component analysis of (log10-transformed) relative abundances of CRC-associated functional categories (as in Fig 4) revealed cancer progression as a dominant source of variation (each dot corresponds to a participant of study population F, color-coded by patient subgroup). The first principal component (PC1) accounting for 43% of the variation in CRC-associated functional changes correlated with cancer progression (Spearman correlation of 0.45, *P* < 1E-8) and stratified non-neoplastic, adenoma, and early- and late-stage CRC patients (color-coded). Shown below are boxplots of each patient's PC1 value grouped by cancer progression. Significant differences between groups were established using pairwise Wilcoxon tests (see bottom right legend). Boxes denote interquartile ranges (IQR) with the median as a black line and whiskers extending up to the most extreme points within 1.5-fold IQR (see Supplementary Fig S6E for a CRC classification based on functional features).

## Discussion

We have shown here that a noninvasive fecal readout allows for accurate detection of CRC in a preclinical setting based on a multi-species predictive model derived from metagenomic data. That both gene and species markers indicate microbiota changes already during early stages of neoplastic growth (Figs 1E, 4 and 5), suggests that identification of reliable microbial markers for advanced adenomas as CRC precursors may be possible. Thus, future studies, ideally in large-scale comparisons to other recently established CRC screening tools (e.g., immunochemical FOBT and (epi-)genetic tests (Allison *et al*, 2007; Hol *et al*, 2010; Imperiale *et al*, 2014)), should systematically explore this possibility because the detection of pre-cancerous lesions with high sensitivity is still a big challenge.

Our results indicate functional and taxonomic associations with CRC and conceptually establish the possibility of CRC detection from fecal microbial markers. Their future application in mass screening will depend on the development of cost-effective assays. Toward this goal, we assessed whether 16S sequencing of fecal samples would be a suitable alternative. A 16S-based classifier for CRC detection, cross-validated on the subset of study population F for which we had

also generated 16S data (*N* = 116), achieved almost as good an accuracy (AUC 0.82; Supplementary Fig S10) as the models based on metagenomic community profiles (AUC 0.84–0.87; Supplementary Fig S10). While our work was under review, another study (Zackular *et al*, 2014) independently arrived at the conclusion that 16S sequencing of fecal samples would allow for accurate CRC screening based on classification models that showed some overlap with our CRC markers at higher taxonomic ranks. But because these were trained on a smaller dataset and were neither cross-validated nor validated in external datasets, it is questionable whether the reported accuracy would be maintainable in other study populations.

While our results of 16S-based CRC detection underline the potential of microbiota for cancer screening, for future development of economic and robust PCR-based assays, it remains to be seen how the 16S gene compares to metagenomics-derived marker genes (Mende *et al*, 2013). Moreover, additional metagenomic data will enable more detailed investigation of cancer-associated differences in gene function, gene content, and genomic variation (Schloissnig *et al*, 2013), serving as a starting point for proposing testable mechanistic hypotheses about the roles of microbiota in cancer onset and progression. This might not only advance our understanding of CRC etiology, but also help to fully realize the potential of the gut microbiome for screening, mortality reduction, and prevention (Faivre *et al*, 2004; Assistance Publique - Hôpitaux de Paris, http://clinicaltrials.gov/ct2/show/study/NCT01270360).

## Materials and Methods

### Data collection for study population F

#### Patient recruitment and characterization at the Creteil Henri Mondor Hospital (France)

Participants of study population F were selected from a cohort of 648 patients recruited with informed consent between 2004 and 2006 from different endoscopy departments at Assistance Publique - Hôpitaux de Paris (academic hospitals) where they had been referred for colonoscopy (detailed in Sobhani *et al*, 2011). The study protocol was approved by the Comité Consultatif de Protection des Personnes dans la Recherche Biomédicale (CCPPRB Créteil-Henri Mondor) that authorized the enrollment of patients in all associated centers and by the respective institutional review board (EMBL Bioethics Internal Advisory Board) and is in agreement with the WMA Declaration of Helsinki and the Department of Health and Human Services Belmont Report. This study population only included participants without previous colon or rectal surgery, colorectal cancer, inflammatory or infectious injuries of the intestine; patients with need for emergency colonoscopy were also excluded. Participants performed a fecal occult blood test (FOBT) at home and sent them via mail to the laboratory (Le centre national de lecture des Hémoccult, Caisse Primaire d'Assurance Maladie (CPAM), Paris) following the standard process of FOBT mass screening in France. They also agreed to give blood for DNA extraction and the *wif-1* methylation assay (as described in Mansour & Sobhani (2009)).

#### Fecal sample collection

Fresh stool samples were collected 2 weeks to 3 days before colonoscopy and in all cases prior to bowel cleanse (Sobhani *et al*,

2011). Whole fresh stool was collected in sterile boxes, and 10 g was frozen at −20°C within 4 h and deposited at the Henri Mondor Hospital biobank CRB (Biological Resources Center). 156 samples were then selected for DNA extraction and shotgun sequencing, among them samples from 53 patients with CRC, 42 adenoma patients, and 61 randomly chosen controls. Colonic neoplasia status was determined by colonoscopy (Table 1, Supplementary Table S1 and Supplementary Dataset S1 for participant metadata). For a subset of 129 patients, 16S rDNA could be amplified from the same DNA extracts and subjected to 16S amplicon sequencing (see below and Supplementary Dataset S2).

## Data collection for study population G

### Patient recruitment at the University Hospital Heidelberg (Germany)
38 colorectal cancer patients without a medical history of inflammatory disease were selected from the ColoCare Study (study population G) (see Supplementary Table S1 and Supplementary Dataset S1 for patient metadata). ColoCare is an international prospective cohort study recruiting newly diagnosed colorectal cancer patients prior to surgery in Germany and the USA. Recruitment sites in Heidelberg are the Department of Surgery at the University Hospital Heidelberg and the affiliated Hospital Salem. Written informed consent was obtained from all study participants. The study protocol was approved by the ethics committee of the Medical Faculty at the University of Heidelberg and by the respective institutional review board (EMBL Bioethics Internal Advisory Board) and is in agreement with the WMA Declaration of Helsinki and the Department of Health and Human Services Belmont Report.

### Fecal sample collection
Fecal samples were collected between diagnosis of colorectal cancer and surgery at least 10 days after bowel cleanse and colonoscopy. Stool samples were stored in RNAlater Solution (Sigma-Aldrich) and frozen at −80°C upon arrival in the laboratory.

## Data collection for study population H

### Fecal sample collection from healthy German participants
Informed consent was obtained from five healthy individuals living in Germany through the my.microbes project (http://my.microbes.eu) to obtain samples as additional controls. The study protocol was approved by the respective institutional review board (EMBL Bioethics Internal Advisory Board) and is in agreement with the WMA Declaration of Helsinki and the Department of Health and Human Services Belmont Report. Fecal samples were collected and conserved under anaerobic conditions in a sealed bag, kept at −20°C for short-term storage, and stored at −80°C upon arrival in the laboratory (see Supplementary Table S1 and Supplementary Dataset S1 for participant metadata).

### Inclusion of published fecal metagenomes
Samples from Danish and Spanish individuals not diagnosed with colorectal cancer were included in study population H as additional controls (Qin *et al*, 2010; Le Chatelier *et al*, 2013). Because they did not undergo colonoscopy, we cannot rule out the possibility that some of them have adenomas or carcinomas, but this is expected to only lead to a slight (conservative) overestimation of false-positive CRC predictions in study population H.

### Fecal metagenomic data from IBD patients
In addition to study population H, we investigated 25 published fecal metagenomes of patients with an inflammatory bowel disease (IBD) (Qin *et al*, 2010) (see Fig 2), but these were not used as controls for CRC classification in study population H (see Supplementary Dataset S1 for participant metadata).

## Data collection for 16S rRNA gene sequencing from colonic tissue

### Patient recruitment at the University Hospital Heidelberg (Germany)
Colonic tissue was collected from 48 patients undergoing colorectal cancer surgery at the Department of Surgery, University Hospital Heidelberg after obtaining informed consent. Written consent procedure and the study protocol were approved by the ethics committee of the Medical Faculty at the University of Heidelberg (see Supplementary Dataset S2 for participant metadata).

### Tissue samples for 16S rRNA gene sequencing
Matched tissue samples (tumor and nearby morphologically healthy mucosa) were collected from these 48 patients within ~30 min after surgical resection, immediately snap-frozen, and stored at −80°C until DNA extraction and 16S rRNA gene sequencing. Of the 48 patients (study population G*), 13 patients also participated in the ColoCare study (study population G) and we not only obtained tissue samples, but also fecal samples in RNALater (Supplementary Dataset S2).

### DNA extraction from stool and tissue samples
Genomic DNA was extracted from frozen or RNAlater-preserved fecal samples as previously described (Furet *et al*, 2009) using the GNOME DNA Isolation Kit (MP Biomedicals) with the following minor modifications: cell lysis/denaturation was performed (30 min, 55°C) before protease digestion was carried out overnight (55°C), and RNAse digestion (50 μl, 30 min, 55°C) was performed after mechanical lysis. Tissue samples were thawed carefully, cleaned from remaining feces with sterile Dulbecco's PBS (PAA Laboratories) if necessary, and the blades were cleaned with Incidin Plus (Ecolab) between samples. DNA was extracted from the tissue surfaces using the protocol above. After final precipitation, the DNA was resuspended in TE buffer and stored at −20°C for further analysis.

## Sequencing and quality control

### Library preparation for metagenomic sequencing
Library preparation was automated and adapted on a Biomek FXp Dual Hybrid, with high-density layout adaptors, orbital shaker, static peltier, shaking peltier (Beckman Coulter, Brea, USA), and a robotic PCR cycler (Biometra, Göttingen, Germany). Magnetic beads were separated on a 96-ring magnet. PCRs were performed in full-skirted plates and sealed with arched auto-sealing lids (Bio Rad, Hercules, USA). Library quality was analyzed on a Fragment Analyzer (Advanced Analytics Technologies, Ames, USA). One-milliliter 96-well microtiter plates were

used as cooling reservoirs for stock solutions containing the enzymes. Bead processing was performed in Abgene 1.2-ml square well u-bottomed plates (Thermo Scientific, Waltham, USA).

Sequencing libraries were generated with SPRIworks HT chemicals (Beckman Coulter) according to the supplier's recommendation with the following modifications: 150 ng DNA starting amount, adaptor dilution 1:25, kit chemical dilution 1:1 in process. Purification steps until PCR were performed according to the protocol (Fisher *et al*, 2011). Using a double size selection we enriched for 250 base fragments, subsequently beads were washed twice with 70% ethanol, and process mixing was performed on the orbital shaker. All additions of reactants were done with span 8 head and sample processing with the 96 tip head. Dead volume and pipetting profile corrections were set with respect to the tip types and solutions used.

### Metagenomic sequencing

Whole-genome shotgun sequencing of fecal samples collected in France and Germany was carried out on the Illumina HiSeq 2000/2500 (Illumina, San Diego, USA) platform. All samples were paired-end sequenced with 100-bp read length at the Genomics Core Facility, European Molecular Biology Laboratory, Heidelberg, to a targeted sequencing depth of 5 Gbp (see Supplementary Dataset S1 for sequencing results).

### 16S rRNA gene sequencing

DNA from 48 tissue sample pairs (tumor and healthy mucosa) and 129 fecal samples (which are a subset of study population F) was amplified using primers targeting the V4 region of the 16S rRNA gene (F515 5′-GTGCCAGCMGCCGCGGTAA-3′, R806 5′-GGACTACHVGGGTWTCTAAT-3′) (Caporaso *et al*, 2011). PCR was carried out according to the manufacturer's instructions of the Q5 high-fidelity polymerase (New England BioLabs, Ipswich, USA) using bar-coded primers (NEXTflex™ 16S V4 Amplicon-Seq Kit, Bioo Scientific, Austin, Texas, USA) at final concentrations of 0.2 μM and an annealing temperature of 56°C for 35 cycles.

PCR products were cleaned up with Agencourt AMPure XP-PCR Purification system (Beckman Coulter, Brea, USA), quantified according to the NEXTflex™ 16S V4 Amplicon-Seq Kit protocol, and multiplexed at equal concentration. Sequencing was performed using a 250-bp paired-end sequencing protocol on the Illumina MiSeq platform (Illumina, San Diego, USA) at the Genomics Core Facility, European Molecular Biology Laboratory, Heidelberg.

## Data analysis

### Taxonomic profiling of fecal samples

Using MOCAT (option *screen* with alignment length cutoff 45 and minimum 97% sequence identity), taxonomic relative abundance profiles were generated by mapping screened high-quality reads (see below for details) from each metagenome to a database consisting of 10 universal single-copy marker genes extracted from 3,496 NCBI reference genomes (Mende *et al*, 2013; Sunagawa *et al*, 2013). Quantification proceeded in two steps, by first estimating taxonomic abundances from all sequenced DNA fragments (nucleotide counts; each read contributing number of counts equal to its length) that

mapped uniquely, and in a second step, nucleotide counts from reads mapping to multiple taxa with the same alignment score were distributed among them proportionally to nucleotide counts originating from reads uniquely mapped to these taxa. Finally, base counts were gene length-normalized (option profiling). Abundance estimates at species level were made based on a recently proposed consistent species-level clustering (Mende *et al*, 2013), while abundance summarization at higher taxonomic levels was based on the NCBI Taxonomy (Supplementary Figs S2 and S3).

### Taxonomic profiling of tissue and fecal samples using 16S rRNA gene sequences

Raw sequencing data were quality-controlled as described below for the generation of the metagenomic gene catalog (minimum read length = 45 bp; minimum base quality score = 20). Paired-end reads were merged using the SeqPrep software (https://github.com/jstjohn/SeqPrep) requiring perfect overlap between the high-quality paired-end reads.

After excluding reads shorter than 200 nt or with more than five ambiguous bases, we aligned them to the SILVA 16S reference database of bacterial and archaeal 16S rRNA sequences. These alignments were cropped to only retain the region spanning V3 to V4 using mothur, version 1.30.2 (Schloss *et al*, 2009). With the same screening and alignment routines, we reanalyzed 16S rRNA sequencing reads from Kostic *et al* (2012) to facilitate comparisons across datasets. For further analysis, we only retained sample pairs with at least 1,000 reads aligned in the tumor as well as the normal tissue sample (a requirement met by 79 sample pairs from Kostic *et al* (2012)). The boundaries of this core alignment were adjusted to accommodate both the pyrosequencing reads from (Kostic *et al*, 2012) and ours from the MiSeq platform, by first separately determining the region spanned by 90% of the reads from each sequencing platform and second by interval merging to cover both of these regions.

For high-resolution taxonomic profiling, we built OTUs from large collections of published 16S rRNA genes. We included 436,028 sequences from the SILVA database (version 115 (Pruesse *et al*, 2007)) and 5,224 extracted from the prokaryotic genome sequences used for taxonomic profiling of metagenomic reads (see above). These were aligned with the same protocol as the 16S reads and cropped to the core alignment region using mothur. Subsequently, we removed redundancy by dereplication (mothur's unique.seqs) and clustering at 98% sequence identity using UCLUST (version 6.0.307 with the option -maxrejects 1000 for more accurate matching) (Edgar, 2010). As an initial quantification and ordering for the heuristic UCLUST algorithm, we mapped 10,000 reads from each sample against the reference sequences and sorted them according to the number of mapped reads using USEARCH routines (98% ID, best-hit).

After we had clustered reference sequences into 2,460 OTUs (with a minimum size of 10), we mapped all 16S Illumina and pyrosequencing reads against these taking a best-hit approach with a minimum of 98% sequence identity between matches (using USEARCH with default settings).

### Generation of the metagenomic gene catalog

Raw paired-end Illumina FastQ files from metagenomic samples were processed using MOCAT (version 1.2) (Kultima *et al*, 2012),

by first removing low-quality reads (option *read_trim_filter* with length cutoff 45 and quality cutoff 20). Retained high-quality (HQ) reads were screened against a custom-made fasta file containing Illumina adapters (option *screen_fastafile* with e-value 0.00001 using USEARCH (Edgar, 2010) version 5). Adapter-screened reads were screened against the human genome version 19 (option *screen* with alignment length cutoff 45 and minimum 90% sequence identity). Screened HQ reads were assembled (option *assembly* with SOAP (version 1.06) (Li *et al*, 2008) and minimum length 500) and the assemblies revised (option *assembly_revision* with minimum length 500). Genes were predicted on the revised scaftigs (option *gene_prediction* using *MetaGeneMark* (Zhu *et al*, 2010) version 2.8).

To obtain a comprehensive metagenomic gene catalog of the human gut microbiome, the predicted genes from this study were pooled with predicted genes from other published human metagenome studies (Qin *et al*, 2010, 2012; Human Microbiome Project Consortium, 2012) and clustered at 95% sequence identity using CD-HIT (Li & Godzik, 2006) (version 4.6.1) (parameters: -c 0.95, -M 0, -G 0 -aS 0.9, -g 1, -r 1, -d 0). The representative genes of each cluster were selected and extended up to 100 bp ('padded region') at each end of the gene by using the sequence information from the assembled scaftigs (Sunagawa *et al*, 2013).

### Functional annotation of the metagenomic gene catalog

The metagenomic gene catalog was annotated by aligning the translated amino acid sequence of each gene to the KEGG (Kanehisa *et al*, 2008) (version 62) ortholog database using BLAST (Altschul *et al*, 1990) (version 2.2.24) (max. e-value 0.01) and then annotating the genes using SmashCommunity (Arumugam *et al*, 2010) (version 1.6). CAZy (Cantarel *et al*, 2009) annotations were made using the dbCAN pipeline (Yin *et al*, 2012) with the recommended default settings; additional substrate information for CAZy families was obtained from CAZypedia (http://www.cazypedia.org/index.php?title=Carbohydrate-binding_modules&oldid=9411, assessed 28 October 2013) and from Cantarel *et al* (2012).

### Functional profiling

Gene abundance profiles were calculated using MOCAT (Kultima *et al*, 2012) by first mapping screened HQ reads from each metagenome to the metagenomic gene catalog (option *screen* with alignment length cutoff 45 and minimum 95% sequence identity). Each gene's abundance was estimated as gene length-normalized nucleotide counts of all reads that matched the protein-coding region of this gene. For each functional feature, its abundance in the metagenomic gene pool was estimated as the sum of the relative abundances of all genes belonging to this family (Fig 4).

### Relative abundance transformations and abundance filtering

For all subsequent analyses, read counts were transformed into relative abundances (by dividing through the total number of reads per sample, including high-quality reads that could not be mapped to reference databases or annotated).

For display purposes and fold change calculations, we applied a logarithmic transformation to relative abundances using the function $\log_{10}(x + x_0)$, where $x$ are the original relative abundances and $x_0$ a small constant (1E-6 for taxonomic and 1E-8 for functional features).

As an unsupervised feature reduction technique (that is independent of any participant metadata), we applied a low-abundance filter, which discarded functional and taxonomic features whose relative abundance did not exceed 0.0001 and 0.001, respectively, in any sample (for combined taxonomic and functional features, we used an abundance cutoff of 0.0001).

### Analysis of diversity and community structure

Species abundances (using the species delineation from Mende *et al* (2013)) were used to calculate Shannon diversity index and species richness for each sample in study population F using the *diversity* and *specnumber* functions, respectively, of the vegan R package (http://cran.r-project.org/web/packages/vegan/index.html). Differences between tumor-free and CRC patients were assessed by the Kruskal–Wallis test (Supplementary Fig S1D and E).

Gene richness (the number of genes from the metagenomic gene catalog with nonzero abundance) was calculated for each sample from study population F after rarefying to 3 million reads per sample; differences were evaluated using the Kruskal–Wallis test (Supplementary Fig S1F).

As an additional high-level descriptor of gut microbial community composition, we analyzed the abundance ratio between the phyla of Bacteroidetes and Firmicutes (Turnbaugh *et al*, 2006) with respect to separation of the three groups of participants using the Kruskal–Wallis test (Supplementary Fig S1C).

Enterotypes were determined on a reference set of the 292 healthy individuals from study population H (Qin *et al*, 2010; Le Chatelier *et al*, 2013) using the original computational protocol and PCoA visualization (Supplementary Fig S1A) (for details, see Arumugam *et al* (2014, 2011)). We projected the 156 samples from study population F into this PCoA space (Trosset & Priebe, 2006) and assigned enterotypes by minimal JSD distance to the medoid of each enterotype (i.e., to the nearest cluster center). Differences in enterotype composition between CRC patients (all stages) and tumor-free controls (some with adenomas) of study population F were assessed using the Fisher test (Supplementary Fig S1B).

Additionally, we subjected study population F to a PCoA independently of other datasets and investigated the separation of CRC cases from controls (neoplasia-free participants and patients with small adenomas) along principal coordinates; significance was assessed using the Wilcoxon test (Supplementary Fig S1G–J).

To assess whether differences in such high-level descriptors of microbial community structure are useful for CRC detection, we built a logistic regression model with the ten first principal coordinates (from Supplementary Fig S1G) and the Bacteroidetes to Firmicutes ratio (Supplementary Fig S1C) as predictors. Its accuracy was determined using tenfold cross-validation on study population F and ROC analysis (Supplementary Fig S1K).

### Confounder assessment

We assessed differences in patient metadata, such as age, gender, and body mass index (BMI), as well as in sequencing depth between CRC cases and tumor-free controls using the Wilcoxon test. While patient age significantly differed between groups, the other variables assessed are unlikely to confound our analyses (Supplementary Fig S5A–D). To determine how predictive the age bias and other variations in patient characteristics are of CRC in our study populations, we built a logistic regression model with patient gender, age, and

BMI as predictors. Its accuracy was assessed in tenfold cross-validation on study population F and in external validation on study populations G and H and compared to the metagenomic model with significance assessed using one-sided DeLong tests (see below and Supplementary Fig S5E). See also below for robustness analysis of the metagenomic classifier against age bias.

### Statistical analysis of differentially abundant taxa and gene functions

To detect significant differences in relative abundances of metagenomic features, we applied the nonparametric Wilcoxon test (or the Kruskal–Wallis test which is its generalization to > 2 groups) as it makes only minimal assumptions about data distributions, which are not well understood for metagenomic data.

We applied this test to compare taxonomic abundance profiles (after abundance filtering, see above) between CRC patients (all stages) adenoma patients and neoplasia-free participants. After false discovery rate correction, features with an adjusted $P$-value < 0.1 were deemed significant (see Supplementary Fig S2).

We additionally compared taxonomic abundances between CRC patients and a control group consisting of neoplasia-free participants and patients with small adenomas using the same approach (Supplementary Fig S3).

For functional analysis, we only included features with an adjusted $P$-value < 0.01 and additionally applied a minimum absolute fold change criterion (> 1.33) to focus on larger, likely more biologically meaningful effects (Fig 4). Here, fold change was defined as the difference between groupwise medians of log-transformed relative abundances (Fig 4).

### Statistical modeling and marker extraction

To distinguish CRC patients from tumor-free controls based on the taxonomic composition of their fecal metagenomes by means of a classifier that extracts the most discriminative features (microbial markers) and to obtain an unbiased measure of its accuracy, we developed a custom pipeline in R (http://www.R-project.org, version 2.12.0). Here, we used the LASSO logistic regression classifier (Tibshirani, 1996) implemented in LIBLINEAR (Fan *et al*, 2008), because it generates a parsimonious classification model, which selects only few features out of a potentially very large set, and because model interpretation and marker extraction is easy due to its linearity, an advantage over, for example, random forests (c.f. Knights *et al*, 2011; Papa *et al*, 2012). Since the feature selection process is built into the LASSO classifier, it is straightforward to obtain not only a parsimonious model, but also a realistic estimate of its generalization error in cross-validation (avoiding a common mistake of dubious two-stage approaches, where supervised feature selection is done before, and independent of, cross-validation, which can lead to dramatically overoptimistic accuracy estimates (see e.g., Smialowski *et al*, 2010).

Our pipeline proceeds as follows:

1 Unsupervised feature abundance filtering to remove extremely low abundant taxa (see above).
2 Feature transformation: We applied the above-described log-transform and subsequently standardized features (by centering to mean 0 and dividing by each features standard deviation to which we added the $10^{th}$ percentile of standard deviations across all features).

3 Partitioning data for tenfold stratified cross-validation (we resampled dataset partitions ten times to obtain more stable accuracy estimates).
4 Fitting a LASSO model on the training data of each cross-validation fold: The LASSO hyperparameter was optimized for each model in a nested fivefold cross-validation on the training subset using the area under the precision–recall curve as model selection criterion and also enforcing at least five nonzero coefficients. To obtain high-precision models, we reweighted examples by assigning the controls five times as much weight as the cases.
5 Application of the trained LASSO models to obtain the corresponding cross-validation test predictions (Fig 1A shows mean predictions from the ten respective test subsets of each sample). Due to the resampled cross-validation (and also in external validation), there are several test predictions for each test examples. To get a single prediction score per example (e.g., as shown in Fig 1A and D and Fig 2A), we averaged all test predictions (from ten or 100 models in cross-validation or external validation, respectively).
6 Model evaluation using ROC analysis: From ten times resampled tenfold cross-validation, we obtained mean test prediction scores, which we subjected to ROC analysis (see Fig 1B and C).
7 Model interpretation and marker extraction: Features (bacterial species) with potential as CRC biomarkers were extracted as nonzero coefficients from all 100 LASSO models (trained in ten times resampled tenfold cross-validation). Fig 1A displays all features that have a nonzero coefficient in at least 50% of the LASSO models in the order of their mean percentage of total absolute coefficient weight across all models. Bar lengths in Fig 1A directly correspond to mean log-odds ratios across LASSO models.

This procedure was used to train and cross-validate the CRC classifier on study population F (Fig 1B) with patients grouped into CRC cases (all stages) and tumor-free controls (including patients with small, but not large adenomas) as label.

A second classifier was trained on the combined study populations F and G (Supplementary Fig S6C), resulting in a more comprehensive and accurate model (see Supplementary Fig S5G for markers present in both models). As a combination test with the FOBT, we trained a third classifier (on study population F) which did not only use microbial species abundance features but also the FOBT results (standardized 0/1 values) as an additional predictor (see Fig 1B; Supplementary Fig S5G for accuracy and feature overlap with the metagenomic classifier).

To assess the effect of taxonomic resolution on CRC detection accuracy in study population F, we reran the described classification pipeline for taxonomic abundance profiles summarized at the genus and phylum level (Supplementary Fig S6D).

Similarly, the predictivity of functional metagenomic features (relative abundances of KEGG modules and CAZy gene families for study population F) was assessed using the same modeling pipeline (Supplementary Fig S6E). Finally, we also built and cross-validated such a model based on a combination of species, KEGG modules and CAZy families as input features on study population F (Supplementary Fig S6E and G).

To explore the effect of sequencing method on the ability to detect CRC, we also applied the classification pipeline to 16S OTU abundance profiles from the subset of study population F for which we had obtained fecal 16S sequencing data (41 CRC cases and 75 controls with small adenomas or without any neoplasia; Supplementary Fig S10).

*External model validation, model comparison and confounder analysis*
Independent (holdout/external) validation consisted of two steps: (1) application of the feature filtering and normalization using the same parameters as for cross-validation data (i.e., discarding features according to the cross-validation low-abundance filter and applying the log-standardization with mean and standard deviation values as estimated in cross-validation, see above) and (2) application of the trained LASSO models to make predictions on the validation data.

As an independent validation of the metagenomic CRC classifier, we applied the models trained on study population F to study population G and H (fitted in cross-validation, see above). From their mean test prediction scores, we determined the ROC curve (Fig 1C). Additionally, we assessed its sensitivity and specificity separately on study populations G and H, respectively (see Supplementary Fig S6A and B).

Confidence intervals for ROC curves (Fig 1B and C) were calculated using the pROC R package (Robin *et al*, 2011). Statistical significance of differences in ROC curves was also assessed using the *roc.test(…, method = 'delong')* from this package (Fig 1B, Supplementary Figs S5E, S6D and E). The significance of differences in TPR at the same FPR cutoff was determined using the bootstrapping subroutine from the same function (Fig 1B).

To rule out that the metagenomic CRC classifier exploits patient age (or BMI) as an indirect predictor of CRC in study population F (in which CRC patients are on average older than controls, see Supplementary Fig S5B), we assessed whether its prediction scores were biased. In case the classifier was confounded by age, one would expect higher prediction scores for samples from older participants. However, we could neither observe an enrichment of old participants among false-positive predictions compared to true negatives, nor among true positives compared to false negatives in any of the study populations F, G, and H (Supplementary Fig S5F). Note that the latter two were not included in the classifier's cross-validation set. Likewise, we ruled out potential confounding by BMI (Supplementary Fig S5G).

*Comparison between fecal metagenomes and 16S biopsy samples*
To establish correspondence between species profiled in fecal metagenomes and OTUs of the 16S rRNA genes from fecal and tissue samples, we first collected genomic 16S rRNA genes for the metagenomic CRC marker species. For each marker species, fragments of the 16S genes corresponding to the 16S read alignments were then extracted and compared to the 16S OTUs using USEARCH (Edgar, 2010) to find the best-matching OTUs within ≥ 97% sequence identity (Note that this approach did not yield a match for marker species for which we could not identify a genomic 16S sequence, or whose 16S gene fragments were too dissimilar from OTU centroids, i.e., more than 3% diverged). The relative abundance and prevalence (defined here as the proportion of samples in which the relative abundance of a species or OTU

exceeded 1E-5) of these OTUs in fecal, tumor, and normal tissue samples was then compared to the relative abundance and prevalence of the corresponding metagenomic fecal marker species in CRC patients and tumor-free controls (Fig 3, Supplementary Fig S9A).

To more globally assess similarities between microbiota at the tumor relative to nearby normal tissue and fecal taxonomic profiles of CRC patients relative to those of tumor-free controls, we compared microbial composition at the genus level. For 16S samples, the genus identity of each OTU was inferred using the RDP classifier (Wang *et al*, 2007), while for metagenomic stool samples, a reference-based approach was taken (see above) (Kultima *et al*, 2012). Genera (after low-abundance filtering, see above) with significant differential abundance between tumor and normal tissue or CRC patients and controls (FDR-corrected $P < 0.1$ and abundance fold change > 5) were determined separately for each datasets (141 fecal metagenomics samples from study population F, 116 fecal 16S samples, which are part of study population F, 48 16S biopsy sample pairs from this study, and 79 16S biopsy sample pairs from Kostic *et al* (2012); see Table 1, Supplementary Table S1 and Supplementary Dataset S1) using unpaired and paired Wilcoxon tests for fecal samples and tumor–normal tissue pairs, respectively. Based on the log-transformed relative abundances of the union set of differentially abundant genera from any of the datasets, we conducted a joint PCA of all samples (Supplementary Fig S9B). Although sample origin and technical differences between 16S and shotgun metagenomic sequencing were detectable in this PCA, it revealed a separation between tumor and control samples along the first principal component that was common to all the datasets analyzed (Supplementary Fig S9B).

*Cancer progression analysis using PCA of CRC-associated functional features*
To reveal dominant trends in functional alterations between CRC patients and tumor-free participants of study population F, we carried out PCA using as input the relative abundances of significantly changing functional features (Fig 5). Correlation between the first principal component (PC1) and cancer progression encoded as four ordered groups of participants with: (1) no neoplasia, (2) adenoma(s), (3) CRC of AJCC stage 0 to II, and (4) CRC of stage III and IV was established by Spearman's rank correlation test and significant differences of PC1 values between these groups by pairwise Wilcoxon tests (Fig 5).

*Targeted analysis of cancer-related gene functions and toxins*
To explore whether genes encoding known bacterial toxins might be enriched in the metagenomes from CRC patients, we collected protein sequences of toxins that have previously been implicated in intestinal diseases, mainly colorectal cancer, from the literature (Fasano, 2002; Dutilh *et al*, 2013). With these, we performed BLAST (Altschul *et al*, 1990) searches against NCBI nr as well as an in-house database of 3,496 high-quality bacterial reference genome sequences and manually selected additional bona fide members for each toxin gene family. Subsequently, we aligned the sequences from each toxin family using Clustal Omega (Sievers *et al*, 2011) and built HMM sequence profiles from these alignments using HMMer 3.0 (Eddy, 2011). E-value cutoffs for HMM prediction were optimized on protein sequences from the in-house database of

reference genomes. Afterward we searched the metagenomic gene catalog (see above) with the profile HMM for each toxin and quantified the abundance of matching sequences in participants of study population F (using the above-described MOCAT routines). Statistical significance was established using the Wilcoxon test (Supplementary Table S2).

### Data availability

The shotgun metagenomic sequencing data and the 16S rRNA amplicon sequencing data from this study are available from the European Nucleotide Archive (ENA) database (http://www.e-bi.ac.uk/ena): accession number ERP005534. Taxonomic abundance profiles derived from metagenomics data are provided as Supplementary Dataset S3.

Published metagenomics datasets analyzed here are available from ENA: accession number ERA000116 (Qin *et al*, 2010) and ERP003612 (Le Chatelier *et al*, 2013). Patient data are provided in Supplementary Datasets S1 and S2.

**Supplementary information** for this article is available online: http://msb.embopress.org

### Acknowledgements

We are thankful to Wolfgang Huber, Athanasios Typas, Christian Widmer, Gunnar Rätsch, and members of the Bork group for inspiring discussions. Additionally, we thank Yan Ping Yuan and the EMBL Information Technology Core Facility for support with high-performance computing as well as Bettina Haase, Dinko Pavlinic, and Bianka Baying and the EMBL Genomics Core Facility for sequencing support. We acknowledge support with patient recruitment and characterization by Gilles Gatineau, Michel Azizi, Thomas Aparicio, Marjan Djabbari, Yann Le Baleur, Isabelle Baumgartner, Hervé Hagège, Anne Courillon-Mallet, Laurent Costes, Robert Benamouzig, Michael Levy, Maryan Cavicchi, Elie Zrihen, Alain Rozenbaum, Gilles Trojmann, Damien Levoir, Olivier Pecriaux, Christophe Lochet, Marc Parieto, Karen Leroy, Bijan Ghaleh, Philippe Le Corvoisier, Mariane Mozer, Philippe Capelle, Benoit Coffin, Marc Gornet, Clare Abbenhardt, Verena Widmer, Stephanie Tosic, Christine Fink, Manja Ghajar Rahimi, Torsten Kölsch, Judith Kammer, and Janina Biazeck and technical support by Karine Le Roux, Sonia Garrigou, Julie Villemot, Sylvia Verdy, Philippe Langella, Jean Pierre Furet, Joël Doré, Jean Pierre Carrau, Lin Zielske, Renate Skatula, Anett Brendel, Hicham Mansour, Biba Nebbad, and Amine Amoura. This work has received funding through the CancerBiome project (European Research Council project reference 268985), the METACAR-DIS project (FP7-HEALTH-2012-INNOVATION-I-305312), the International Human Microbiome Standards project (HEALTH-F4-2010-261376), a French LNCC 2004 grant, ACD grants, and in part through the PHRC 2009 Vatnimad grant.

### Author contributions

PB, IS, JT, MvKD, AYV, GZ, and SS conceived and managed the project. AYV, AA, FB, NH, MKo, AL, MS, PS-K, JB, CT, JTVN, MKl, CMU, MvKD, and IS recruited and diagnosed patients and collected samples. AYV, JT, RH, JZ, and VB generated sequencing data. GZ, JT, AYV, SS, JRK, PB, PIC, DRM, and TY designed and performed data analysis. GZ, PB, AYV, JT, SS, and JRK wrote the manuscript with contributions from all other authors.

### Conflict of interest

The authors declare that they have no conflict of interest.

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
