## [Review Process File · Molecular Systems Biology]

Potential of fecal microbiota for early stage detection of colorectal cancer

Georg Zeller, Julien Tap, Miss Anita Voigt, Shinichi Sunagawa, Mr. Jens Roat Kultima, AurÉlien Amiot, Francesco Brunetti, Mr. Paul I Costea, Nina Habermann, Miss Rajna Hercog, Moritz Koch, Alain Luciani, Mr. Daniel Mende, Martin A. Schneider, Petra Schrotz-King, Christophe Tournigand, Jeanne Tran Van Nhieu, Takuji Yamada, Juergen Zimmermann, Vladimir Benes, Matthias Kloor, Cornelia M. Ulrich, Magnus von Knebel Doeberitz, Iradj Sobhani, Peer Bork

Corresponding author: Peer Bork, EMBL

Review timeline:

Submission date:	01 August 2014
Editorial Decision:	02 September 2014
Revision received:	27 October 2014
Editorial Decision:	07 November 2014
Revision received:	12 November 2014
Accepted:	13 November 2014

Editor: Thomas Lemberger

Transaction Report:

1st Editorial Decision

02 September 2014

Thank you again for submitting your work to Molecular Systems Biology. We have now heard back from two of the three referees who agreed to evaluate your manuscript. Given that their recommendations are similar, I prefer to make a decision now rather than delaying the process further. As you will see from the reports below, the referees are cautiously supportive. They raise, however, several concerns on your work, which should be convincingly addressed.

Without repeating all the points raised by the reviewers, two major issues are the following:

- an analysis separating the three groups--normal subject vs subjects with adenomas vs subjects with carcinoma--should be provided to at least report the observed trends across these groups.
- in view of the recent study by Schloss et al, a side-by-side comparison of the results obtained with metagenomic data and with the available 16S rRNA data, respectively, should be provided.

On a more editorial level, please add a 'data availability' section at the end of the Materials & Methods section to specify where the patient-level metagenomic, FOBT and wif1 methylation data can be accessed. We would also suggest that you provide the source data that include the measurements visualized in Figure 1A.

If you feel you can satisfactorily deal with these points and those listed by the referees, you may

wish to submit a revised version of your manuscript. Please attach a covering letter giving details of the way in which you have handled each of the points raised by the referees. A revised manuscript will be once again subject to review and you probably understand that we can give you no guarantee at this stage that the eventual outcome will be favorable.

Reviewer #1:

The manuscript by Zeller and colleagues represents a large amount of impressive work attempting to predict the presence of colon cancer based on the metagenomic content found in stool. I have several critical concerns with the current manuscript. First, it is implicitly assumed by the authors that shotgun data is better than 16S when they have the data to test the hypothesis. Second, the normal and adenoma samples are pooled to compare the pooled samples against carcinoma subjects. This is an odd choice from a clinical perspective. Third, no hypothesis tests are performed to test whether one model is better than another. The necessary corrections would require major changes to the manuscript.

1. I realize that the authors submitted their manuscript in the last month, but since then Patrick Schloss's research group published a paper using 16S rRNA gene sequencing to differentiate subjects who were normal, had adenomas, and had carcinomas (<http://www.ncbi.nlm.nih.gov/pubmed/25104642>). They also included the subjects' clinical and demographic metadata including the result of an FOBT test. Much of their modeling approach overlaps with what is described here. In addition, they had an earlier study with a similar approach to predict the presence of *C. difficile* (<http://www.ncbi.nlm.nih.gov/pubmed/24803517>). I don't hold these studies against the current study; however, the authors do need to comment on these. Specifically, it is important to note why one would need to go through the considerable effort of performing shotgun metagenomic sequencing when similar/better results can be obtained using amplicon data. The authors also need to include patient demographic/clinical data to see whether the shotgun data provides a statistically significant improvement in the ability to categorize subjects.

2. Many of the studies the authors list in their introduction ascribing a role for the microbiome are based on animal and tissue culture-based models. For example, the enterotoxigenic *Bacteroides fragilis* story has yet to be validated in humans.

3. The beginning of the Results section includes an interesting analysis of the data using enterotypes. Unfortunately, there are several problems with this analysis. First, since enterotype definitions are based on the samples being included, I wonder why the authors did not re-cluster the samples with the references. Second, they indicate that the "distribution of enterotypes varied slightly between these groups". A p-value is needed to support this sentence. Third, the PAM-based approach to defining enterotypes has been shown to be flawed by researchers like Chris Quince, who have proposed a more robust method based on Dirichlet Multinomial Mixture models. I wonder why the data have not been reanalyzed using this approach. This re-assessment is important because samples with large numbers of Fusobacteria and Proteobacteria were not used to define the original set of enterotypes.

4. I have a significant concern that the authors have pooled the normal subjects with those having adenomas in an effort to differentiate them from carcinomas. The authors state: "...to distinguish CRC patients from tumor-free controls (including adenoma patients)..." My reading of this sentence is that the normals and adenomas were pooled. I don't understand why. Clinically, it is very important (and difficult) to detect adenomas and relatively easy to detect carcinomas using other methods. Also, the goal of detecting carcinomas is problematic, since early detection is critical to the subjects' success.

5. Throughout the manuscript there are a number of statements made about their models that lack an assessment of whether the model is significantly better than other approaches. The authors need to re-evaluate all of their claims. One example is the conclusion that the AUC was 0.83 using metagenomic data. They need to indicate a p-value for whether that AUC is significantly better (or the sensitivity/specificity) than the FOBT or methylation rates.

6. This manuscript is being sold as a classifier that could be used to diagnose individuals. Yet, rarely would one only have their metagenome. I wonder why the authors have not included the subject's sex, age, FOBT, or methylation results. Again, this would help make the paragraph starting "As the age distribution differs between cases and controls..." more robust. P-values will be critical to support the value of adding metagenomic data.

7. The validation experiments are a strength of this study.

8. The authors need to include the Crohn's disease paper published by Xavier when discussing the role of *Fusobacterium* in IBD (<http://www.ncbi.nlm.nih.gov/pubmed/24629344>).

9. Since the authors have the DNA sequences and the DNA itself, were they able to identify a relationship with ETBF or the *E. coli* pks genes?

10. I would like the authors to defend the choice of using shotgun sequence data to build a classifier. Yes, metagenomic sequence data is useful for getting to potential functions and possibly to strain level assignments. However, I am surprised that the AUC values described here aren't much different from those presented in the study from the Schloss lab that is described above. The authors did generate the 16S rRNA data from the same samples as their metagenomes. It would be interesting for them to generate models based on the 16S rRNA data to see if it is as good as the metagenomic data. Of course, this will require a statistical hypothesis test. I feel that metagenomic shotgun data is being used based on the false premise (or at least not well supported premise) that it is better than 16S for performing an analysis based on taxonomy.

11. Incidentally, I am not blown away by the metabolic information that is gained from having the shotgun sequence data. Many of the vague results (mucin, pro-inflammatory) could have been made by just looking at 16S rRNA gene sequences. Furthermore, just because the genes are present does not necessarily mean that they're being expressed. The authors need to be more careful about how these data are being presented.

Reviewer #2:

Summary

The authors describe metagenomic sequencing of 156 fecal samples from France (103 healthy controls and 53 CRC patients) to assess the suitability of the fecal metagenome as a predictor for CRC. They find significant changes between patients and controls particularly on the species level. The most consistently increased species reported were two *Fusobacterium* spp., *Porphyromonas asaccharolytica* and *Peptostreptococcus stomatis*. Employing the abundance of multiple bacterial species instead of one marker species in fecal samples they present a non-invasive mean to detect CRC which is more accurate than previously applied methods in clinical settings (fecal occult blood test (FOBT) and Wif-1 gene methylation test). Since the three analysed tests were shown to be at least partially independent of each other, a combination of them (or at least the FOBT and the bacterial-abundance method) could increase the true positive rate from 49% to 66% with an AUC of 0.86.

The applicability of the presented approach was also investigated on a previously published metagenome of 297 fecal samples from Germany and Denmark to assess any regional bias. This study population only consisted of individuals not diagnosed with cancer and inflammatory bowel disease and the false positive rate was 5.1% (in comparison to 3.9% in the French population). The slight increase could be explained by the fact that the Danish/German population was not assessed with colonoscopy (and thus false-negatives could be present).

Additionally, the presented method was validated on fecal samples from 38 German individuals with diagnosed CRC. A combination of the French and German cohorts improved the AUC from 0.83 to 0.86 (only bacterial marker species).

Due to the importance of early detection of CRC the authors also tested the applicability of their approach for detection of early stage cancers and observed a comparable or even slightly higher sensitivity for the detection of early stage tumours than for late stage tumours. The abundance of *Fusobacterium* spp. and *Peptostreptococcus* spp. was particularly high in early stage cancers.

The across-disease specificity of the bacterial CRC markers was assessed by applying the method to a small metagenome from 21 UC and 4 CD patients. Even though the false positive rate increased to

16%, indicating at least partially similar changes in the microbiota of IBD and CRC patients, most of the observed changes in the microbiota were reported to be specific to CRC.

The observed changes in the fecal metagenome were also observed in the metagenome of 127 tumor-normal tissue pairs, in particular for the potential marker species *Fusobacterium* spp., *Peptostreptococcus stomatis*, *Eubacterium* spp. and *Streptococcus salivarius*.

Analysis of the function of the CRC associated microbiome revealed a preference for genes involved in mucin degradation among others in the fecal CRC metagenomes. This finding could reflect on the possible degradation of mucin by CRC associated, cell-adhesive or invasive, bacteria such as *Fusobacterium* spp. More speculatively, the increase in polyamine metabolism (incl. putrescine) was linked with the increase of bacterial species which were originally discovered as oral pathogens (e.g. *Fusobacterium* spp, *Porphyromonas* and *Peptostreptococcus*) and the fact that putrescine/spermidine metabolism was described as a core trait of the oral microbiota. Lastly, a correlation between CRC microbiota gene signature and the progression of the disease was reported.

Critical view of the study

The major finding of the study was that multiple bacterial species could be applied to detect CRC in a relatively large cohort (491 individuals in 3 cohorts) and that the presented method was comparable or better than currently used methods for early, non-invasive CRC detection.

The data presented supports the key conclusion and is based on a reasonably large cohort. One of the limitations of the presented method, the disease specificity of the reported microbial marker species, was at least considered in a small cohort of IBD patients.

Previous studies only reported the overabundance of various OTUs in luminal and mucosal material from CRC patients in comparisons to controls but did not systematically investigate the potential of the fecal microbiota as a predictor for CRC. Thus Zeller et al. present a novel finding. To the best of my knowledge, the only other study systematically assessing the fecal microbiota (Zackular et al., 2014; with very similar findings to the here reviewed study) was published during the review process and is thus excluded as a criterion in this review due to the "Scooping Protection Policy" of Molecular Systems Biology.

The major drawback of the presented study is the fact that whole-metagenome sequencing is as of today not feasible to use as a diagnostic tool for CRC detection in a clinical setting due to methodological and financial limitations. It would have been advantageous if Zeller et al. would have provided 16S-rRNA sequencing data for fecal samples and possibly (q)PCR data targeting the established marker bacteria to demonstrate the reproducibility of their findings with less expensive and less time consuming methods (which could eventually be employed in a clinical setting).

Lastly it is questionable if the presented study fits the scope of Molecular Systems Biology whose primary emphasis is on molecular components and their interactions. Only the functional analysis of the CRC microbiota can lead to speculations regarding the interactions of the microbiota and the human host at the interface of the mucus layer. The rest of the study is a census of the CRC microbiota and its potential application for detection of CRC.

Minor points

- No keywords provided
- Missing bracket: (part of study population F, see Suppl. Table S3 that were subjected to 16S amplicon sequencing
- Change to Fig. 5: "from precancerous stages to metastasized carcinomas (Fig. 4)" and "KEGG modules and CAZy families (as shown in Fig. 4)"

Summary of changes in the revised manuscript:

Before answering the reviewers point by point we summarize the major changes:

- The three groups of participants (neoplasia-free, adenomas and carcinomas) were analyzed in pair-wise comparisons to motivate the inclusion of small adenomas in the control group

for CRC detection

- For all subsequent CRC classification analyses, large adenomas were excluded and CRC patients compared to a control group of neoplasia-free and small-adenoma participants (as has been done in other large-scale CRC screening studies). This caused most Figures (and Expanded View) to change, however without affecting any conclusion.
- The analysis of high-level structural community features (enterotypes, Bacteroides:Firmicutes ratio, principal coordinates) was expanded to show that while some of these differ significantly, they are not highly predictive of CRC.
- The confounder analysis was extended to show that patient metadata in our study populations is not useful for accurate CRC screening and does not improve the metagenomic CRC classifier when used as additional predictors. Some patient data is now shown in Table I.
- The combination test based on metagenomic features and FOBT results is highlighted more and now shown in Figure 1B.
- CRC classification models based on functional features (abundances of KEGG modules and CAZy families) was added and a model based on both functional and taxonomic features indeed slightly improves over the best taxonomic model.
- A CRC classifier trained on 16S OTU data was added. This classifier achieves an AUC > 0.8, which corroborates the conclusion about a strong microbial signal for CRC that can be robustly detected. A brief conceptual comparison of the two sequencing approaches is discussed together with this result.

Editor:

Without repeating all the points raised by the reviewers, two major issues are the following:

- an analysis separating the three groups--normal subject vs subjects with adenomas vs subjects with carcinoma--should be provided to at least report the observed trends across these groups.

We have added a thorough comparison of microbial taxa (phyla, genera and species) whose abundance differs significantly in any of the pair-wise comparisons between the three groups of patients (Figure E2). From the apparent similarity of the microbiota from neoplasia-free and adenoma subjects our rationale to include adenoma patients in the control group should become clear. We have however modified the setup of CRC detection by excluding all large adenoma patients from the cross-validation data, so that the classifiers are trained (and evaluated) to distinguish CRC patients from controls without neoplasia or small adenomas (many of which would never develop into carcinomas), as has been done in another large-scale CRC screening study [Imperiale et al., NEJM, 2014].

- in view of the recent study by Schloss et al, a side-by-side comparison of the results obtained with metagenomic data and with the available 16S rRNA data, respectively, should be provided.

We have added Figure E10 to show that CRC classification based on 16S rRNA data is possible with almost the same accuracy as from metagenomic species profiles. This result is discussed together with a concise conceptual comparison of the two sequencing approaches (see Discussion).

On a more editorial level, please add a 'data availability' section at the end of the Materials & Methods section to specify where the patient-level metagenomic, FOBT and wif1 methylation data can be accessed. We would also suggest that you provide the source data that include the measurements visualized in Figure 1A.

We have added a 'Data availability' section pointing to ENA where the newly sequenced metagenomes can be obtained. There we also added pointers to the published data sets we used. We have provided taxonomic and functional abundance profiles as additional supplemental data as well as the parameter vectors of the CRC classification models to facilitate reproduction of Figure 1.

Reviewer #1:

The manuscript by Zeller and colleagues represents a large amount of impressive work attempting to predict the presence of colon cancer based on the metagenomic content found in stool. I have several critical concerns with the current manuscript. First, it is implicitly assumed by the authors that shotgun data is better than 16S when they have the data to test the hypothesis. Second, the normal and adenoma samples are pooled to compare the pooled samples against carcinoma subjects. This is an odd choice from a clinical perspective. Third, no hypothesis tests are performed to test whether one model is better than another. The necessary corrections would require major changes to the manuscript.

*1. I realize that the authors submitted their manuscript in the last month, but since then Patrick Schloss's research group published a paper using 16S rRNA gene sequencing to differentiate subjects who were normal, had adenomas, and had carcinomas (<http://www.ncbi.nlm.nih.gov/pubmed/25104642>). They also included the subjects' clinical and demographic metadata including the result of an FOBT test. Much of their modeling approach overlaps with what is described here. In addition, they had an earlier study with a similar approach to predict the presence of *C. difficile* (<http://www.ncbi.nlm.nih.gov/pubmed/24803517>). I don't hold these studies against the current study; however, the authors do need to comment on these. Specifically, it is important to note why one would need to go through the considerable effort of performing shotgun metagenomic sequencing when similar/better results can be obtained using amplicon data. The authors also need to include patient demographic/clinical data to see whether the shotgun data provides a statistically significant improvement in the ability to categorize subjects.*

Even though the study by Zackular et al. (Cancer Prev. Res. 2014, from Pat Schloss's group) arrives at the similar conclusion that fecal microbiota read outs should be useful for CRC screening, we would like to emphasize here that we take a fundamentally different approach to assess the generalization error of our metagenomic CRC classifiers (as most of the following applies to the *C. difficile* study as well, we will not discuss it separately). While the goal of generalizing from the analyzed data set to further (unseen) samples seems like a trivial one, achieving it requires taking statistical precautions against overfitting and confounding (see e.g. <http://en.wikipedia.org/wiki/Overfitting> or Chapter 7 of Hastie, Tibshirani & Friedman, The Elements of Statistical Learning, 2009).

Zackular et al. only report training errors for their classification models as far as we could see. They specifically mention that these models were not cross-validated, nor was an external data set used for model validation, which is in fundamental contrast to our approach (as also noted by the Reviewer below).

Furthermore, in the study by Zackular et al. patient age, gender, ethnicity and BMI apparently differ between patients and controls (in most cases significantly, see their Table 1), which is worrisome since this can confound the analysis of CRC associated microbiota. Fitting a classifier to these training set biases (as done in Zackular et al.) essentially amounts to overfitting them – a particular danger for small data sets. While this can result in a seemingly good model when assessed on the same training data (as shown in Zackular et al.), it will typically generalize poorly to the whole screening population (which does not exhibit similarly strong bias in age, gender, ethnicity and BMI).

In contrast to their study, we carefully checked if, and ruled out that, our CRC classifier exploits an age bias that is present in our study population F (Figure E5) thereby adhering to standard statistical practice. We moreover extended this confounder analysis in the revised manuscript by showing that patient metadata are not useful for CRC screening in our larger study populations (N=156 and N=335 versus N=90 in Zackular et al.): A model only based on patient metadata yields AUCs < 0.7 (Figure E5 E) and when we incorporated patient metadata into the metagenomic classifier, its accuracy did not improve at all.

For a discussion of whether the sequencing approach matters, see below (in response to remark 10).

2. Many of the studies the authors list in their introduction ascribing a role for the microbiome are based on animal and tissue culture-based models. For example, the enterotoxigenic Bacteroides

fragilis story has yet to be validated in humans.

We rephrased the Introduction to clarify that the mechanistic work we are referring to has been carried out in cell lines and mouse models.

3. The beginning of the Results section includes an interesting analysis of the data using enterotypes. Unfortunately, there are several problems with this analysis. First, since enterotype definitions are based on the samples being included, I wonder why the authors did not re-cluster the samples with the references. Second, they indicate that the "distribution of enterotypes varied slightly between these groups". A p-value is needed to support this sentence. Third, the PAM-based approach to defining enterotypes has been shown to be flawed by researchers like Chris Quince, who have proposed a more robust method based on Dirichlet Multinomial Mixture models. I wonder why the data have not been reanalyzed using this approach. This re-assessment is important because samples with large numbers of Fusobacteria and Proteobacteria were not used to define the original set of enterotypes.

Admittedly enterotypes are a concept that is actively debated in the research community and a consensus on methods for assigning enterotypes has not been reached. We feel however that these issues are peripheral to our study. Here our key point is to show that any high-level descriptor of microbiome community structure (more of these are now included in the revised manuscript, see Figure E1) – even if they differ significantly between groups – are not going to be useful for accurate CRC detection. To make this clearer and to address the request of clustering our data set independently, we added several panels to Figure E1 (C, G to K).

A p-value for differences in enterotype distribution was given in the original submission already on top of the corresponding bar plot (in the revision this is Figure E1B).

4. I have a significant concern that the authors have pooled the normal subjects with those having adenomas in an effort to differentiate them from carcinomas. The authors state: "...to distinguish CRC patients from tumor-free controls (including adenoma patients)..." My reading of this sentence is that the normals and adenomas were pooled. I don't understand why. Clinically, it is very important (and difficult) to detect adenomas and relatively easy to detect carcinomas using other methods. Also, the goal of detecting carcinomas is problematic, since early detection is critical to the subjects' success.

The Reviewer is correct to point out that adenomas are more difficult to detect than carcinomas and his concern that the CRC classifier might be biased against the detection of adenomas is not entirely unjustified when these are included in the control group. However it is important to note that, while most adenomas will not develop into cancers, advanced/large adenomas are a strong risk factor. We therefore modified the setup of our CRC classification and excluded patients with large adenomas from the control group for the sake of classifier training and evaluation (as has also been done in Imperiale et al., NEJM 2014).

Additional justification for this choice is presented in Figure E2 showing the results of a three-way comparison between neoplasia-free controls, adenoma patients and CRC patients. The figure clearly shows strong similarity between the microbiota of neoplasia-free controls and adenoma patients (just a single significant difference detected), whereas many of the statistically significant differences between neoplasia-free controls and CRC patients are also seen in a comparison between adenomas and CRC. We modified the manuscript to include this motivation of our choice to include the small adenoma patients (but not those with large adenomas) in the control group.

Due to this change of the classification setup, Figures 1-3 (and essentially all Supplemental Material) had to be redone. However none of the central conclusions were affected.

5. Throughout the manuscript there are a number of statements made about their models that lack an assessment of whether the model is significantly better than other approaches. The authors need to re-evaluate all of their claims. One example is the conclusion that the AUC was 0.83 using metagenomic data. They need to indicate a p-value for whether that AUC is significantly better (or the sensitivity/specificity) than the FOBT or methylation rates.

Although we are convinced that a difference in AUC justifies statements about whether one method is more accurate than another, we agree that it can be informative to assess the statistical

significance of these differences as a means of quantifying their uncertainty with respect to the particular data set used. We have added these analyses to the revised manuscript and adjusted the decision boundary of the metagenomic classifier to match the specificity of FOBT. Figure 1 now clearly shows that a combination test, which uses metagenomic features and FOBT results, is significantly more accurate than either method alone (Figure 1).

6. This manuscript is being sold as a classifier that could be used to diagnose individuals. Yet, rarely would one only have their metagenome. I wonder why the authors have not included the subject's sex, age, FOBT, or methylation results. Again, this would help make the paragraph starting "As the age distribution differs between cases and controls..." more robust. P-values will be critical to support the value of adding metagenomic data.

See our answers above to remark 1 and 5.

7. The validation experiments are a strength of this study.

We are grateful for this remark (see also remark 1 above).

8. The authors need to include the Crohn's disease paper published by Xavier when discussing the role of Fusobacterium in IBD (<http://www.ncbi.nlm.nih.gov/pubmed/24629344>).

We included a reference and also mentioned the finding from this study that Fusobacterium was found to be enriched in mucosal tissue biopsies, but not in stool of Crohn's patients, which might explain why we did not observe a strong Fusobacterium enrichment in fecal samples from IBD patients we analyzed.

9. Since the authors have the DNA sequences and the DNA itself, were they able to identify a relationship with ETBF or the E. coli pks genes?

The answer to these and similar questions about bacterial virulence factors previously suspected to play a role in CRC could be found in the original Supplement (in the revised manuscript Table E2) and was/is discussed in the last paragraphs of the Results section (we now explicitly mention the negative results for *BFT* and *pks* on page 13).

10. I would like the authors to defend the choice of using shotgun sequence data to build a classifier. Yes, metagenomic sequence data is useful for getting to potential functions and possibly to strain level assignments. However, I am surprised that the AUC values described here aren't much different from those presented in the study from the Schloss lab that is described above. The authors did generate the 16S rRNA data from the same samples as their metagenomes. It would be interesting for them to generate models based on the 16S rRNA data to see if it is as good as the metagenomic data. Of course, this will require a statistical hypothesis test. I feel that metagenomic shotgun data is being used based on the false premise (or at least not well supported premise) that it is better than 16S for performing an analysis based on taxonomy.

On the comparability of accuracy estimates between our study and that from the Schloss lab we have commented above (see remark 1). Because of the issues explained above, we do not perceive the main difference between these two studies to be in sequencing approaches. While shotgun metagenomic data clearly contains much more information than amplicon data (see revised Discussion), we are also convinced that there is broad consensus in the field about the more limited taxonomic resolution/accuracy of amplicon data (and that there often is a bit of a taxonomic identification problem of the actual OTUs, as can be seen in Zackular et al and in our Figure E10 B). Despite these challenges in data interpretation, we clarified in the revised manuscript that 16S sequencing might be a viable approach for fecal microbiota-based CRC screening (Discussion). This is essentially suggested by an additional classifier trained on 98% OTUs from the subset of 116 fecal samples of study population F for which we also had generated amplicon data. This classifier achieved an AUC of 0.82, which is not much less than the best comparable metagenomics models (AUC 0.84-0.87; Figure E10). The main conclusion we draw from this is that the microbial associations with CRC are so strong, even in fecal samples, that accurate CRC detection is robust to differences in the actual read-out technology.

11. Incidentally, I am not blown away by the metabolic information that is gained from having the shotgun sequence data. Many of the vague results (mucin, pro-inflammatory) could have been made by just looking at 16S rRNA gene sequences. Furthermore, just because the genes are present does not necessarily mean that they're being expressed. The authors need to be more careful about how these data are being presented.

The reviewer seems to suggest that the functional potential (e.g. genetic potential for mucus degradation or pro-inflammatory metabolism) could be inferred from 16S sequencing data. Although metagenome reconstruction methods from 16S data have recently been proposed (e.g., Langille et al. Nat. Biotechnol. 2013), their resolution is limited (as is also acknowledged in that reference). In particular pathogenicity, genotoxicity and induction of host inflammation are good examples of processes that are very difficult to infer from any taxonomic profile, because virulent and non-virulent organisms, such as enterotoxigenic and non-toxigenic *Bacteroides fragilis* strains or *E. coli* strains with and without the *pks* genomic island as a trigger of colonic inflammation (see e.g. Arthur et al. Science 2012), can be so closely related that they are indistinguishable in terms of their 16S rRNA sequences.

Similarly mucolytic capabilities of gut microbiota are only partially understood to date (and whether overrepresentation of particular mucolytic species are beneficial or detrimental to the host is even less clear; see e.g. Png et al. Am. J. Gastroenterol. 2010 or El Katouri et al., Nat. Rev. Microbiol. 2013). To our knowledge, CAZy analysis of (meta-)genomes (as done here) is considered best practice for inferring the metabolic capability for host glycan utilization (e.g., El Katouri et al., Nat. Rev. Microbiol. 2013).

We are well aware that we can only analyze the abundance of genes encoding certain functions, which clearly is not a direct readout of functional activity. We thus carefully worded the Results sections using the term “enrichment in metagenomes” (in the revised manuscript even more strictly than before) and clearly marked speculative consequences of the observed enrichments as such. We moreover openly discuss potential caveats of our analysis (such as the cause-consequence issue or the fact that we do not have data on dietary preferences of study participants).

Reviewer #2:

Summary

The authors describe metagenomic sequencing of 156 fecal samples from France (103 healthy controls and 53 CRC patients) to assess the suitability of the fecal metagenome as a predictor for CRC. They find significant changes between patients and controls particularly on the species level. The most consistently increased species reported were two *Fusobacterium* spp., *Porphyromonas asaccharolytica* and *Peptostreptococcus stomatis*. Employing the abundance of multiple bacterial species instead of one marker species in fecal samples they present a non-invasive mean to detect CRC which is more accurate than previously applied methods in clinical settings (fecal occult blood test (FOBT) and *Wif-1* gene methylation test). Since the three analysed tests were shown to be at least partially independent of each other, a combination of them (or at least the FOBT and the bacterial-abundance method) could increase the true positive rate from 49% to 66% with an AUC of 0.86.

The applicability of the presented approach was also investigated on a previously published metagenome of 297 fecal samples from Germany and Denmark to assess any regional bias. This study population only consisted of individuals not diagnosed with cancer and inflammatory bowel disease and the false positive rate was 5.1% (in comparison to 3.9% in the French population). The slight increase could be explained by the fact that the Danish/German population was not assessed with colonoscopy (and thus false-negatives could be present).

Additionally, the presented method was validated on fecal samples from 38 German individuals with diagnosed CRC. A combination of the French and German cohorts improved the AUC from 0.83 to 0.86 (only bacterial marker species).

Due to the importance of early detection of CRC the authors also tested the applicability of their approach for detection of early stage cancers and observed a comparable or even slightly higher sensitivity for the detection of early stage tumours than for late stage tumours. The abundance of *Fusobacterium* spp. and *Peptostreptococcus* spp. was particularly high in early stage cancers. The across-disease specificity of the bacterial CRC markers was assessed by applying the method to a small metagenome from 21 UC and 4 CD patients. Even though the false positive rate increased to 16%, indicating at least partially similar changes in the microbiota of IBD and CRC patients, most

of the observed changes in the microbiota were reported to be specific to CRC. The observed changes in the fecal metagenome were also observed in the metagenome of 127 tumor-normal tissue pairs, in particular for the potential marker species Fusobacterium spp., Peptostreptococcus stomatis, Eubacterium spp. and Streptococcus salivarius.

Analysis of the function of the CRC associated microbiome revealed a preference for genes involved in mucin degradation among others in the fecal CRC metagenomes. This finding could reflect on the possible degradation of mucin by CRC associated, cell-adhesive or invasive, bacteria such as Fusobacterium spp. More speculatively, the increase in polyamine metabolism (incl. putrescine) was linked with the increase of bacterial species which were originally discovered as oral pathogens (e.g. Fusobacterium spp, Porphyromonas and Peptostreptococcus) and the fact that putrescine/spermidine metabolism was described as a core trait of the oral microbiota. Lastly, a correlation between CRC microbiota gene signature and the progression of the disease was reported.

Critical view of the study

The major finding of the study was that multiple bacterial species could be applied to detect CRC in a relatively large cohort (491 individuals in 3 cohorts) and that the presented method was comparable or better than currently used methods for early, non-invasive CRC detection.

The data presented supports the key conclusion and is based on a reasonably large cohort. One of the limitations of the presented method, the disease specificity of the reported microbial marker species, was at least considered in a small cohort of IBD patients.

Previous studies only reported the overabundance of various OTUs in luminal and mucosal material from CRC patients in comparisons to controls but did not systematically investigate the potential of the fecal microbiota as a predictor for CRC. Thus Zeller et al. present a novel finding. To the best of my knowledge, the only other study systematically assessing the fecal microbiota (Zackular et al., 2014; with very similar findings to the here reviewed study) was published during the review process and is thus excluded as a criterion in this review due to the "Scooping Protection Policy" of Molecular Systems Biology.

For extensive comments on the study by Zackular et al. we kindly refer you to our answers to Reviewer #1 above.

The major drawback of the presented study is the fact that whole-metagenome sequencing is as of today not feasible to use as a diagnostic tool for CRC detection in a clinical setting due to methodological and financial limitations. It would have been advantageous if Zeller et al. would have provided 16S-rRNA sequencing data for fecal samples and possibly (q)PCR data targeting the established marker bacteria to demonstrate the reproducibility of their findings with less expensive and less time consuming methods (which could eventually be employed in a clinical setting).

We agree that shotgun metagenomics is not a suitable technology for clinical application of our approach. While we tried our best to make clear that we present a conceptual advance here that is not yet fully developed into a clinical diagnostics, we have added a classifier that based on 16S OTU abundance data can detect CRC with only slightly reduced accuracy (Figure E10). In terms of assay cost, a 16S based test may already be competitive to new CRC diagnostics, such as the Cologuard test that costs about \$600 [see also Imperiale et al, NEJM 2014]. For the sake of robustness and time to result, we nevertheless believe that qPCR readouts should be explored in the future, but we feel that this clearly is beyond the scope of this study.

Lastly it is questionable if the presented study fits the scope of Molecular Systems Biology whose primary emphasis is on molecular components and their interactions. Only the functional analysis of the CRC microbiota can lead to speculations regarding the interactions of the microbiota and the human host at the interface of the mucus layer. The rest of the study is a census of the CRC microbiota and its potential application for detection of CRC.

We added some more analyses based on functional metagenomic profiles to the revised manuscript (Figure E6 E) and would also point to the analysis of virulence factors previously associated with CRC (last paragraphs of the Results section). Ultimately however, we feel that it is an editorial decision whether our study does or does not fit the scope of the journal.

Minor points

- No keywords provided

We have included them in the revised manuscript.

- *Missing bracket: (part of study population F, see Suppl. Table S3 that were subjected to 16S amplicon sequencing*

Thank you. Fixed in the revised manuscript.

- *Change to Fig. 5: "from precancerous stages to metastasized carcinomas (Fig. 4)" and "KEGG modules and CAZy families (as shown in Fig. 4"*

Our intention was to point to Figure 4 here, because the heatmap quite clearly visualizes the transition (color gradient) from neoplasia free controls in the left column over adenomas in the middle to carcinomas in the rightmost columns apparent for the majority of functional modules and gene families shown.

Reviewer #3

Zeller et al report the identification of taxonomic markers in the stool microbiota that differentiate colorectal cancer patients from healthy controls. The analysis includes the construction of a diagnostic model by LASSO, differential abundance testing of taxonomic and microbial gene markers by Wilcoxon test, and cross-validation within and between cohorts. Results were compared across microbial community assays (16S and metagenomic sequencing) and against clinical FOBT.

The authors convincingly argue that their metagenomic classifier has a remarkable specificity (~5%) and better accuracy than the fecal occult blood test. If replicated in future work, this result is impressive and represents a large step forward for the microbiome community, as it shows that the composition of the microbiome can be a better indicator of the disease state than some current clinical diagnostics. As this work is based on metagenomic sequencing, it is still quite removed from being an affordable clinical diagnostic test itself, but it is a significant step towards that goal.

By showing that even with the more cost-effective 16S rRNA sequencing read-out we can reach almost the same accuracy for CRC detection as with metagenomic profiling (Figure E10), we provide direct evidence in the revised manuscript that cost issues may not be a principle impediment to clinical translation.

Major Comments

The authors use microbial gene function analysis to show an increase in fiber utilization in non-CRC patients and increased utilization of host glycans and amino acid degradation in CRC patients; these are relevant insights into the functional role of the microbiome in colon cancer. Although they are typically higher dimensional than taxonomic markers, CAZy family and KEGG module features are both also potentially more diverse and mechanistically informative than community composition and might better capture the CRC versus healthy state -- why not use these features for building an alternative classifier or improving upon the taxonomy-based classifier?

We thank the reviewer for this suggestion. Indeed, CRC classification based on either KEGG module or CAZy family features does not reach the accuracy of the species model (Figure E6 E). Its accuracy is similar to what we see for classification at higher taxonomic ranks (Figure E6 D) hinting at the possibility that with improved functional annotation methods we might in the future be able to build more predictive functional metagenomic disease classifiers. Already at this point it is apparent that functional metagenomic features capture somewhat orthogonal information to the taxonomic profiles, because a CRC classifier that is trained on both functional and taxonomic features outperforms any taxonomic or functional model (Figure E6 E).

In the enterotype analysis for which the authors observed a slight enrichment of CRC patients in the Bacteroides 'type', would it not be more consistent with the data to use a continuously valued

Bacteroides:Firmicutes ratio, or loading vector from ordination space, as a summary statistic for compositional enrichment? This would avoid the need to discretize the apparently continuous space of community configurations (Supp. Fig. 1) into arbitrary discrete units for this analysis.

In principle we agree that discretizing data is not necessarily suitable as a preprocessing technique for abundance analysis. However, here the key point we tried to make is that all of the high-level descriptors of microbiome community structure (be it enterotypes, ordination axes or the Bacteroidetes:Firmicutes ratio) – even if they differ significantly between groups – are bad choices for CRC classification compared to species abundance profiles (see also our answers to Reviewer #1). To make this clearer, we added 6 panels to Figure E1 (C and G to K) showing that a classification model based on principle coordinates and the Bacteroides:Firmicutes ratio results in much lower CRC detection accuracy.

Minor Comments

Fig4 -- the colors distinguishing enriched/depleted in CRC and the heatmap could be better differentiated

We purposefully chose colors indicating enrichment/depletion in CRC metagenomes that are similar to the heatmap color scheme, as the enrichment/depletion information directly corresponds to fold change in CRC that is visualized in the rightmost columns of the heatmap.

2nd Editorial Decision

07 November 2014

Thank you again for submitting your work to Molecular Systems Biology. We are now globally satisfied with the modifications made and I am pleased to inform you that we will be able to accept your study for publication pending the following final minor edits:

- please clarify in the text that large-adenoma patients were indeed included in the "Adenoma" group used in the pairwise comparison shown in Figure E2.
- please indicate the statistical significance of the "strong overlap ... in the differences between CRC versus neoplasia-free and CRC versus adenomas".

2nd Revision - authors' response

12 November 2014

Thank you for the positive decision on our submission to Molecular Systems Biology. We have made the final edits that you requested.

- We clarified in the main text and caption of Supplementary Fig S2 that the adenoma group includes large ones.
- We indicated in Supplementary Fig S2 which changes are also observed when excluding the large adenomas from the adenoma group.